# Information-based centralization of locomotion in animals and robots

Izaak D. Neveln[1], Amoolya Tirumalai [1] & Simon Sponberg [1,2]

The centralization of locomotor control from weak and local coupling to strong and global is hard to assess outside of particular modeling frameworks. We developed an empirical, model-free measure of centralization that compares information between control signals and both global and local states. A second measure, co-information, quantifies the net redundancy in global and local control. We first validate that our measures predict centralization in simulations of phase-coupled oscillators. We then test how centralization changes with speed in freely running cockroaches. Surprisingly, across all speeds centralization is constant and muscle activity is more informative of the global kinematic state (the averages of all legs) than the local state of that muscle's leg. Finally we use a legged robot to show that mechanical coupling alone can change the centralization of legged locomotion. The results of these systems span a design space of centralization and co-information for biological and robotic systems.

[1] School of Physics, Georgia Institute of Technology, Atlanta, GA, USA. [2] School of Biology, Georgia Institute of Technology, Atlanta, GA, USA. Correspondence and requests for materials should be addressed to I.D.N. (email: ineveln2@gmail.com)

Animal locomotion, the task of actively moving from one position and orientation to another, is achieved through nonlinear dynamics, where control is typically distributed across many components. For effective locomotion, coordination of muscles and limbs in space and time is necessary to produce directed forces. Locomotor coordination could either be achieved through strong, global coupling with dense connections between components, or through weak, local coupling with sparse connections[1]. The continuum between these extreme coupling paradigms is thought of as the centralization/decentralization axis of locomotor control[2]. While the concept of centralization is prevalent, it has been difficult to find a quantifiable measure that can be applied across various systems. For example, Brambilla et al. define a decentralized robotic swarm to consist of autonomous individuals that communicate locally and receive no global information[3]. Cruse et al. define stick insect motor control as decentralized because muscle commands are more modulated by peripheral feedback rather than the central nervous system[4]. However, either strong mechanical coupling or centrally integrated sensory feedback could also result in a highly centralized control architecture[2]. Also, the ability to reduce a complex dynamical system to low-dimensional model while still capturing dynamics of the system, especially under perturbations could indicate a highly centralized architecture[5]. Methods to assess the empirical centralization of locomotor systems, preferably without any assumption of an underlying dynamic model, are necessary to answer questions regarding how a multi-actuated system is coupled through mechanics, feedback, and control.

An example of an unresolved locomotor hypothesis is if centralization tends to increase or decrease with the speed of movements. These questions about centralization have been explored most in insects such as cockroaches and stick insects. For fast movement, control via sensory feedback might be less effective due to inherent time delays and bandwidth limitations[6]. This hypothesis predicts a reliance more on fast decentralized mechanical and neural responses local to each leg[2]. While there is some evidence that neural feedback is too slow to effectively coordinate control for fast locomotion from experiments in cockroaches[7] and flies[8], some examples of fast local sensory feedback exists[9,10]. An alternative hypothesis is that internal feedforward control may need to be highly centralized to maintain dynamic stability at high speeds[11]. There is some evidence that overall coupling increases with speed[12] and that precision in timing of leg movements is coordinated through internal neural coupling[8,11]. However, there is currently no general measure of centralization for a system that does not rely on a specific modeling framework, leaving questions regarding the varying degree of centralization in control of animal movement largely unresolved.

Many model-based measures exist for quantifying the centralization of systems. Given a full network model, node centrality can indicate which nodes in the network most govern information flow[13]. Distributions of node centrality over networks also indicate overall network architecture[14]. Coupled-oscillator network models can exhibit coordinated or synchronized behavior similar to the coordination of neural networks or the mechanics of limbs in animals[15]. The Kuramoto model of many globally phase-coupled oscillators has been well characterized[16], where oscillators transition from endless incoherence to fast synchronization as coupling increases and global influences outweigh local influences[17]. Coupled oscillators have been used to represent networks of central-pattern generator (CPG) circuits that drive leg movements[15,18,19]. These coupled-oscillator models have been used to estimate coupling strengths between control of legs in animal systems[12] as well as controllers for robotic systems[20]. A good centralization measure should track changes in coupling to the oscillator models used to describe dynamical systems.

While increased coupling of CPGs should result in increased centralization, so should increased mechanical coupling and feedback. Systems can be coordinated solely through mechanical coupling, such as a passive walker[21]. Mechanical coupling can also affect mechanosensory feedback circuits that detect changes in force to one leg due to the lift-off of others as has been investigated in stick insects[22]. Interlimb coordination, including energy-efficient gait transitions, can be achieved in a quadruped robot solely through local force sensing without any other communication between the leg controllers[23]. Force changes can affect the ability to coordinate locomotion as seen in flies[24]. A measure of centralization should reflect how shifts in mechanics can change overall coupling whether through changes in the passive dynamics or feedback circuits that depend on mechanical signals.

Here we take an information-based approach to quantifying information assessing how much mutual information a control signal shares with global and local states of the locomoting system. We also separately measure how much net information the control signal shares with both local and global states using a separate quantity, co-information[25,26], which measures net redundancy. The next section introduces the theory behind these centralization and co-information measures. We then validate these measures of centralization and co-information using a coupled-oscillator network of locomotion to ensure that it can reproduce results in a model where centralization has been previously defined as the coupling strength. Next we measure centralization in running cockroaches and test the hypothesis that cockroach control becomes more centralized at faster running speeds. To then test if our centralization measure could detect changes in mechanical coupling alone, we analyze centralization of a mechanically coordinated robot with variable inertia. Finally, we discuss how these various systems map onto an information space containing centralization and co-information and how this space can be used as a tool for comparing biological control strategies as well as designing robotic control strategies.

## Results

**An information-theoretic measure of centralization**. What unifies concepts of centralization is the amount of information a control signal shares about the global state of the system compared to the amount shared with the local state is greater for more centralized systems than the amount shared with the local state (Fig. 1a). Here we develop an empirical measure that can be used to test hypotheses of centralization of control across different systems but is also in agreement with previous models. More information will be shared between a control signal and a global state than with a local state in the centralized system with stronger mechanical coupling, internal connection, and global feedback. We use an information-theoretic approach which can assess the dependencies among varied locomotor signals. This approach does not require modeling the dynamics of how these signals interact. The measure is general in the sense that it can be used regardless of whether the signals are neural spiking patterns, kinematics, voltages or forces and does not depend on the particular relationship between the signals. Also, as the measure always expresses the shared information between the signals using the same units of information, we can make broad comparisons between different systems. Still, the measure of centralization, just as any empirical measure, will necessarily depend on the specific signals used to represent the system. Supplementary Note 1 outlines the steps needed to obtain a reliable estimate of the centralization of a system.

Each of the three signals has some amount of variation, which is quantified by entropy $H$ and can be represented as an

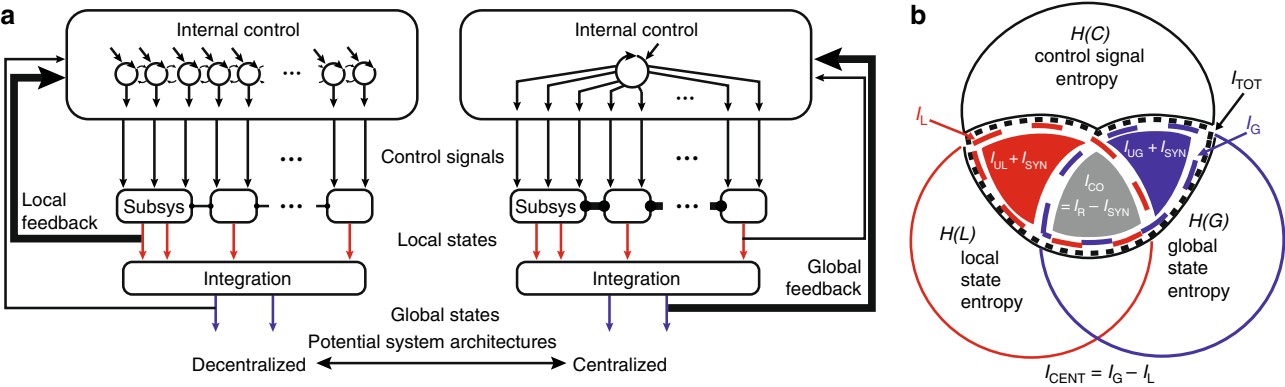

**Fig. 1** Measuring Centralization of Locomotor Systems. **a** Locomoting systems can use numerous possible control signals which cascade information through successive levels of integration to effect a relatively small number of global task related outcomes. Coupling between control can range from weak and local (decentralized) to strong and global (centralized). Arrows indicated direction of information flow. Lines connecting the subsystems represent mechanical coupling, with thickness signifying coupling strength. Centralization could also be affected by the degree of coupling in the internal architecture of control, as well as the strength of global feedback versus local feedback. **b** Our centralization measure uses empirical observations of control signals, local states, and global states to infer the coupling of control. We estimate the mutual information between the control signal and both the global and local states. We measure centralization as $I_G$ minus $I_L$, which removes any $I_R$ between the three variables. We expect decentralized systems to contain more $I_L$ and centralized systems to contain more $I_G$. $I_{UG}$, $I_{UL}$, $I_{SYN}$, and $I_R$ are the unmeasured constituent parts of the total information $I_{TOT}$ shared between the control signal and both local and global outputs

area (Fig. 1b, Supplementary Fig. 1). When two signals have interdependencies, knowing one signal decreases the amount of variation or entropy in the other. Mutual information $I$ measures this decrease in the entropy of one signal when the other signal is known. We measure $I$ between the control signal and the local and global states giving us estimates for local mutual information ($I_L$) and global mutual information ($I_G$). For the mathematical definitions of these information-theoretic measures as well as how they are estimated, see Methods and Supplementary Note 2.

$I_L$ and $I_G$ are not necessarily independent as there could be the same redundant information $I_R$ in both, where knowing something about the control signal lowers entropy in both local and global states (Supplementary Fig. 2). $I_L$ and $I_G$ could also be synergistic, providing more information together than the sum of their individual contributions. Ideally, we would want to estimate how much of the total mutual information $I_{TOT}$ between the control signal and both states is uniquely global ($I_{UG}$) versus uniquely local ($I_{UL}$). The axioms that allow for decomposing these information components are debated, and estimating these quantities is possible yet challenging for real data[25]. We avoid these issues by simply subtracting $I_L$ from $I_G$, thus canceling $I_R$. We can compare the relative amounts of $I_{UL}$ and $I_{UG}$ by defining our measure of centralization to be

$$I_{CENT} = I_G - I_L = (I_{UG} + I_R) - (I_{UL} + I_R)$$
$$= I_{UG} - I_{UL}, \quad (1)$$

where the redundant information contained in both $I_G$ and $I_L$ is eliminated. The values for $I_{CENT}$ could range from $-I_{TOT}$ if all information is uniquely local, to $I_{TOT}$ if all information is uniquely global.

By measuring $I_{TOT}$ we can also derive a common information-theoretic measure called co-information[25,26] ($I_{CO}$) given by

$$I_{CO} = I_L + I_G - I_{TOT} = I_R - I_{SYN}, \quad (2)$$

where $I_{SYN}$ is the amount of synergistic information shared between the control signal and both local and global states only when both are known. $I_{CO}$ is similar to $I_{CENT}$ in that it is a difference of two positive constituents of $I_{TOT}$ and could have the same range of values (Supplementary Fig. 2). $I_{CO}$ is a measure of

net redundancy and does not contain the unique information in local or global states[25]. A negative value indicates that synergistic information outweighs redundant information. $I_{CENT}$ and $I_{CO}$ are therefore two measures of information differences that look at how different parts of the total information are balanced, and both can be potentially useful to discriminate different types of neuromechanical control systems.

Grounding these measures back into specific biological signals, a positive value of $I_{CENT}$ indicates that electromyograph (EMG) activity from a selected muscle is more informative about the global kinematic state averaged from all limbs than the local kinematic state of the leg in which the muscle resides. Also, positive $I_{CENT}$ guarantees nonzero $I_{UG}$, meaning that there must be global information not present locally. Therefore, this global information would have to come from some source of coupling (mechanical or neural) within the system. A positive value for $I_{CO}$ indicates some net redundancy between local and global information. As $I_{CO}$ increases, it becomes less important to know both local and global states to have information about the control signal, whereas a negative value for $I_{CO}$ indicates that knowing both local and global states together gives more information about the control signal.

**Oscillator model centralization rises with coupling strength.** We first test whether our measure of centralization captures the previous model-specific definition of centralization based on the strength of a phase-coupled oscillator network shown in Fig. 2a. Phase-coupled oscillator models are used as a tool to understand locomotion[15], and this particular model has been used to estimate coupling strength between the six legs used in cockroach locomotion[12]. The dynamics of each oscillator phase $\theta_i$ is given by

$$\dot{\theta}_i = 2\pi f_i + \sum_{j=1}^{6} K a_{ij}\sin(\theta_j - \theta_i - \varphi_{ij}) + 2\pi\nu_i + 2\pi P_i, \quad (3)$$

where $f_i$ is the natural frequency of each oscillator (set to 10 Hz to be comparable to cockroach stride frequencies), $a_{ij}$ is 1 if there exists a connection between oscillator $i$ and $j$ and is zero otherwise, $\varphi_{ij}$ is the preferred phase difference between oscillator $i$ and $j$, $\nu_i$ is additive Gaussian noise (0 Hz mean, 0.03 Hz standard

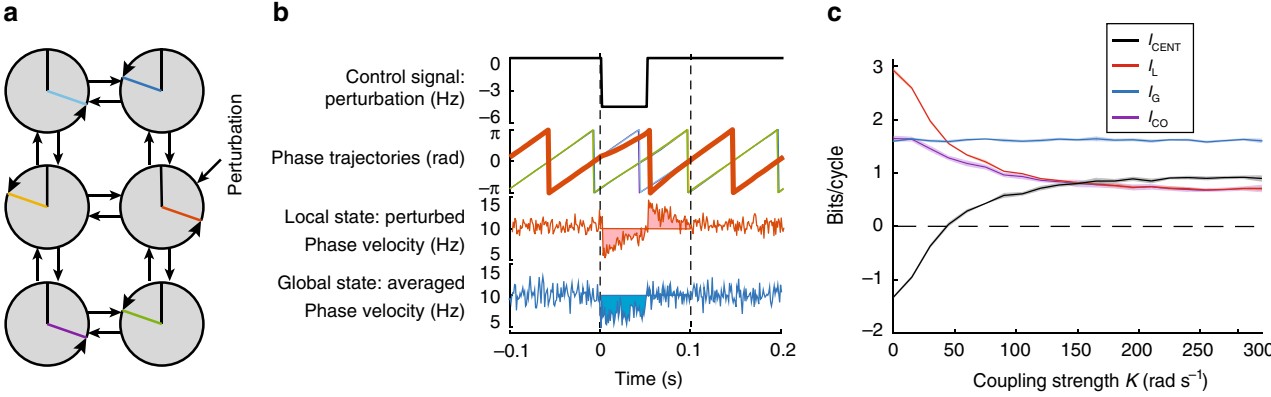

**Fig. 2** Centralization of a coupled-oscillator model of locomotion. **a** We simulated this six phase oscillator network described in Eq. (3) that has been used to model the alternating tripod gait of cockroaches. **b** The control signal was a perturbation to a single oscillator with random amplitude. The local oscillator phase, shown in red, deviates more than other oscillators. The local state was the integrated absolute deviation from baseline of the phase velocity of the perturbed oscillator, shown by the shaded red region. The global state was similarly calculated from the average phase velocity. **c** For a range of coupling strengths $K$ we calculated the mutual information between the pulse height of the perturbation and the local ($I_L$) and global states ($I_L$ and $I_G$, respectively). The shaded regions show the uncertainty in the estimate of mutual information

deviation), and $K$ is the coupling strength between oscillators. The sinusoid coupling term is zero when phases are at the preferred phase difference and drive the phases toward that phase difference otherwise. We integrate Eq. (3) using the Euler–Maruyama method[27].

We want to characterize how the information present in a perturbation to an oscillator is spread throughout the network. We prescribe a square pulse $P_i$ lasting one half cycle put into just one oscillator as shown in Fig. 2b with a pulse height drawn from a Gaussian distribution ($-5$ Hz mean, $\frac{4}{3}$ Hz standard deviation). We then measure both the local response of that oscillator (the integrated deviation away from the steady state phase velocity) to that perturbation and the average global response (same as local only all phase velocities are averaged together[17]) of all oscillators as shown in Fig. 2b. We can then estimate $I_L$ between the perturbation and the local response and $I_G$ between the perturbation and the global response to calculate $I_{CENT}$ and $I_{CO}$.

The model is highly decentralized when $K = 0$ rad s$^{-1}$, where $I_L$ outweighs $I_G$. Also, $I_{CO}$ equals $I_G$ indicating that any $I_G$ is redundant with local information resulting in no unique information represented in the global state $I_{UG}$, as shown in Fig. 2c. The perturbation cannot propagate to the other oscillators due to full decoupling and no additional information can be present in the global signal that is not in the local signal. As coupling is introduced and increases, centralization increases, becomes positive, and levels out to a maximal value. At coupling strengths above $K = 150$ rad s$^{-1}$ the $I_L$ is completely redundant, meaning the value for $I_{CENT}$ equals the amount of $I_{UG}$. Thus, though $I_G$ stays constant with increased coupling strength, $I_{UG}$ must increase from zero to a positive value as a positive value of $I_{CENT}$ requires that there exists $I_{UG}$. Changes to coupling strength can manifest in physical oscillators through changing the mass of a freely moving platform that holds a number of metronomes[28] or increasing the number of connections between central pattern generating circuits driving locomotion[29]. Our centralization measure can empirically infer the relative coupling strengths of these systems, validating it as a useful diagnostic tool. Furthermore, centralization could be used to analyze other types of oscillator models, building on previous work to understand the strength and direction of coupling using information-theoretic analysis[30].

**Cockroaches are centralized during running.** Having validated the measure with a simple model, we next apply it to test biological hypotheses of locomotor control in the running cockroach. We ran 17 cockroaches over flat ground while recording EMG activity from the femoral extensor muscle 137 in the middle leg and tracking the 2D kinematics of the ends of all six legs as shown in Fig. 3a. This muscle has previously been implicated in control even during high-speed running[31]. We collected a wide range of stride frequencies to test centralization across speeds (Fig. 3b). Muscle activity is parameterized by the number of spikes (Fig. 3c) and the timing of those spikes (Fig. 3d). Separating the EMG into spike count and spike timing allows us to test the relative importance of rate versus timing encoding with regards to centralization[32]. We calculated the mutual information between these two muscle parameters and the local (Fig. 3e) and global (Fig. 3f) output states. The trajectories of these states in Fig. 3e, f are colored by the timing of the first spike for the corresponding stride, which indicates some dependency between spike timing and the local and global states. A two-dimensional representation of the output states, where the trajectories were sampled at two phases shown by the dashed lines in Fig. 3e, f, produced similar mutual information estimates as higher dimensional representations (see Supplementary Figs. 3 and 4 and Supplementary Note 3).

When analyzing 2982 strides from all cockroaches (Fig. 3b–f), we find that $I_{CENT}$ is positive (Fig. 3g). Positive $I_{CENT}$ means motor unit spikes are more informative about the global average kinematics than the local kinematics of the limb where the control signal originates. It is surprising that the activation of a muscle in one leg indicates more about the average state of all the limbs than that of the leg it directly activates. This main result is likely because of strong neural and mechanical coupling between the legs[12,33]. Moreover, the positive value for centralization matches that of the coupled-oscillator model for $K > 50$ rad s$^{-1}$, which is in the range of coupling strengths previously fit to the coupled-oscillator model from an earlier cockroach kinematics study[12]. Without having to assume a particular model for the coupling (i.e., phase), our centralization measure recapitulates this earlier results and provides added insight that the global state information is actually larger than the local state information.

A second new finding comes when we split the information into that conveyed by the number of spikes (the count) and the

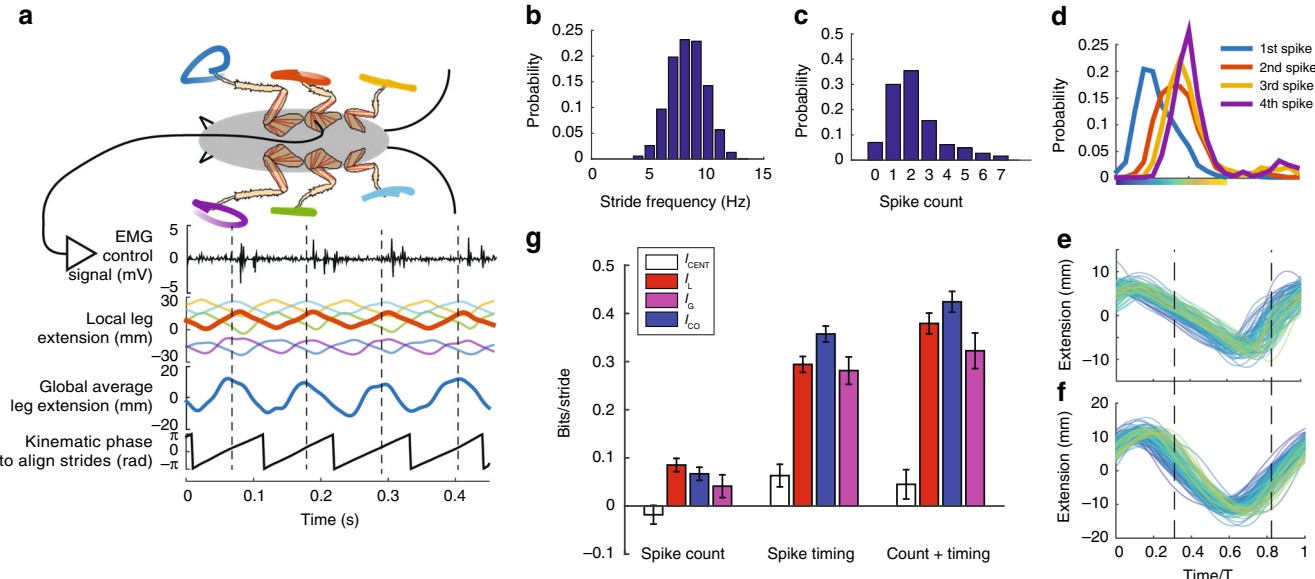

**Fig. 3** Centralization of cockroach locomotion. **a** Control, local, and global signals recorded from the cockroach. Strides were separated according to kinematic phase calculated using the Phaser algorithm[62]. **b** Distribution of stride frequencies across all 2982 strides taken from 17 animals. **c** Distribution of the number of spikes in the femoral extensor over a stride. **d** Probability density functions of the timing of the first four spikes if present in a stride, with time normalized by stride period $T$. **e** Local leg extension trajectories colored by the timing of the first spike (colormap from **d**). Correlations between the timing of the first spike and the resulting local and global states evident by the fact that blue strides with an early spike are distinguished from yellow strides with a late spike. **f** Global leg extension trajectories colored as in **e**. **g** $I_{CENT}$, $I_G$, $I_L$, and $I_{CO}$ for all strides

specific spike timings. There is growing appreciation that precise spike timing codes may play an important role in motor control and not just in sensory encoding[34]. Recent studies show that the timing of individual motor unit spikes has causal effects on motor dynamics down to the millisecond scale in other insects[35] and birds[36]. When just spike count is considered, $I_L$ slightly outweighs $I_G$, though the contribution of overall information is small. Most information, and the positive value for $I_{CENT}$, arises only when spike timing is also considered. Many analyses of motor neuronal activity in insects use only the spike count or rate[37–39]. It is possible that much of the encoded information regarding leg coupling is suppressed in such analyses.

Muscle 137 (as well as its homologous muscle 179 in the hind leg) is driven by a single fast motor neuron that also drives other extensor muscles 136, 135d′, and 135e′ in the middle leg (178, 177d′, and 177e′ in the hind leg)[40]. These muscles can produce varying mechanical work from the same signal[41], including positive work to drive extension or negative work to slow flexion. Therefore, this single motor unit has been implicated in the control of leg flexion and reversal[42] as well as the start of joint extension and stride length[31]. Our results indicate that the control signal for the middle leg shares non-redundant information with both the stance (extending) and swing (flexing) portions of the stride, which both corroborates the reported versatility of this motor unit as well as the observation that muscle work depends on the state of the limb[43].

**Speed affects mutual information but not centralization.** Given that the cockroaches tested exhibited a wide range of stride frequencies (Fig. 3b), we next tested whether faster speeds were more centralized possibly for maintaining dynamic stability[11] or more decentralized possibly due to bandwidth constraints[2]. When we segment the cockroach data into slow and fast halves according to stride frequency, we observe that $I_{CENT}$ does not change (leftmost column of Fig. 4). However, when considered on

a per stride basis, both $I_G$ and $I_L$ are higher for slower strides than for faster strides (Fig. 4a). Information in spike timing is the main contributor to this trend. When just spike count is considered, $I_G$ and $I_L$ are slightly lower in the slower group, though again count information contributes much less information overall. When converted to bits per second using the median frequency of each group (Fig. 4b), we actually see that the information per unit time (bit rate) is greater for the faster group.

Though the balance of local and global information does not change, perhaps the two states become more redundant with greater speed. Overall, $I_{CO}$ per stride is similar between fast and slow groups. However, the faster group is closer to full redundancy as the $I_{TOT}$ is smaller. The slower group has higher $I_{CO}$ in spike timing and lower $I_{CO}$ in spike count. Timing is therefore more redundant for the slow group, whereas $I_{CO}$ is actually negative in count indicating some degree of synergy between local and global states. Therefore, for the slow strides, both output states together share more information with the number of spikes in a stride than when taken separately.

Due to bandwidth constraints and delays, faster strides are expected to have diminished ability for control[7]. The decreases in both $I_L$ and $I_G$ per stride are evidence supporting this prediction. However, the decrease per stride is not as much as expected if one assumes a constant information rate, as faster strides carry more information per unit time. The predictions for centralization are complicated because while the decrease in information with speed is expected, internal and mechanical coupling are hypothesized to increase to maintain dynamic stability[44]. Spatial coordination[8] and temporal coordination[38,45] have been shown to increase with speed[8]. When fitting bursts of activity from the cockroach's thoracic ganglia to coupled-oscillator models, no correlation was observed between burst frequency and coupling strength[46]. Only a very weak positive correlation between running speed and coupling strength was observed when fitting free-running cockroach leg kinematics to such a model[12]. Our measure of centralization, which takes into account a neural control signal

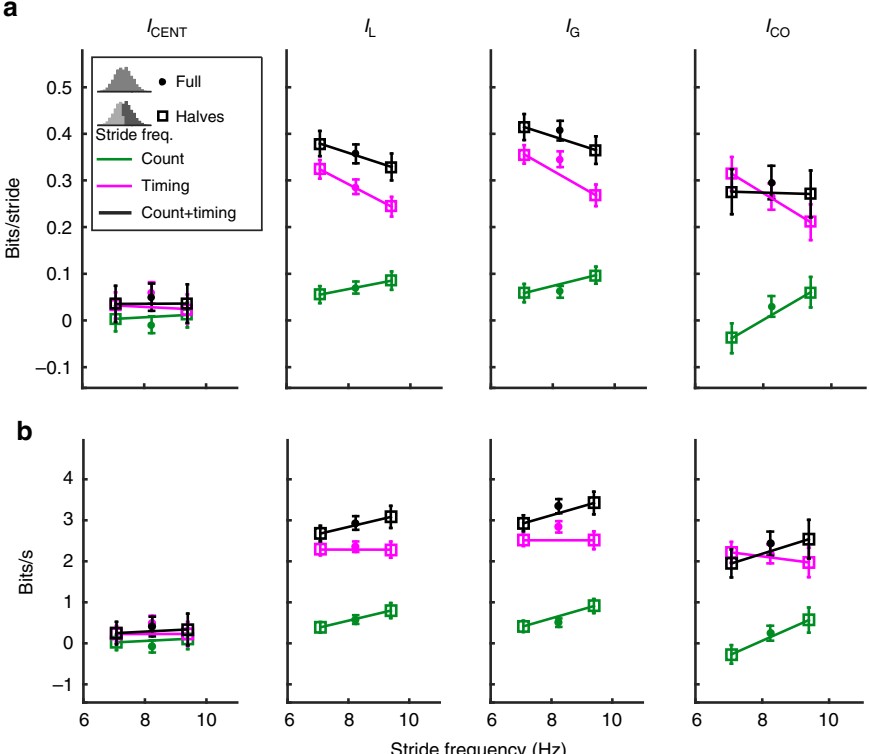

**Fig. 4** Centralization and of fast versus slow cockroach strides. **a** When splitting the strides into a slow half and a fast half, $I_{CENT}$ remains unchanged when compared to the full set of strides. However, both local and global mutual information decrease equally, and this decrease is attributed to information in timing. $I_{CO}$ remains constant due to competing trends in timing and count. **b** Same as **a** only information is converted to a bits/second rate using the median stride frequency of each group. Though the information per stride decreases, the information per unit time increases

with local and global kinematics indicates no shifts in overall coupling with speed when considering cockroach running, though we note that they are overall centralized according to our measure, whereas we might predict that slower insects such as stick insects might be more decentralized[4].

**Robot coordination via mechanical coupling is decentralized**. If neural feedback delays are too long to effectively couple limbs during fast locomotion to properly respond to perturbations, mechanical coupling could potentially compensate. Furthermore, changes to mechanical coupling can alter feedback signals related to the state of the system and its parts[23]. Clearly, mechanics must be considered when analyzing the control architecture of dynamic locomotion. We test whether our empirical measure of centralization can detect changes to mechanical coupling. The Minitaur robot (Ghost Robotics, Inc. Philadelphia, PA) shown in Fig. 5a demonstrates coordination through mechanical coupling[47]. As one pair of legs impacts the ground, the rest of the body translates and rotates in the sagittal plane, generally resulting in movement of the hips of the alternate leg pair. This movement is paramaterized by $\kappa = i_b/m_b d^2$, where $i_b$ is the rotational inertia around the pitch axis, $m_b$ is the mass of the robot, and $d$ is half of the hip-to-hip distance. Even if the commanded torque to one leg pair does not explicitly depend on the states of the other leg pair (i.e., no internal "neural" coupling), the two leg pairs will tend toward a bound gait where the front pair of synchronized legs is antiphase with the synchronized back pair when $\kappa < 1$, or a pronk gait where all legs are synchronized when $\kappa > 1$. Transitions between these gaits can occur by changing mechanical coupling through changes to the moment of inertia $i_b$

around the pitch axis, or by adding phase coupling into the internal control[47].

We altered $\kappa$ by shifting two weights in opposite directions longitudinally along the robot to change $i_b$. We expected $I_L$ to be greater than $I_G$ with the largest difference near the decoupled mode at $\kappa \approx 1$. We also predicted $I_{CO}$ would be positive and close to the value of $I_G$, because any information transfer through mechanical coupling should be redundant if the system is relatively stiff.

We ran the bound gait described in ref. [47] over flat ground, which still produces variability in each stride. We measured local mutual information between the axial force estimated and the actual extension of that leg as shown in Fig. 5b. We compare the local information to the global mutual information between that same torque signal and the average extension trajectories of all four legs (see Supplementary Note 3 for more detail).

$I_L$ is greater than $I_G$ for all experimental conditions, resulting in a negative value of $I_{CENT}$. $I_{CENT}$ is minimized for the intermediate $\kappa$ condition for the rear leg pair, confirming the prediction for when mechanical coupling is minimized. $I_{CENT}$ is greatest for the low $\kappa$, where $I_G$ is fully redundant. For the front leg pair, $I_{CENT}$ is minimized for both the intermediate and high $\kappa$ conditions.

$I_{CENT}$ of control for signals from the front legs is higher when considering the front leg versus the rear leg, indicating an asymmetry to the mechanical coupling not predicted in the reduced models of the Minitaur that only consider bounding in place[47]. This asymmetry is the same regardless of which of the front or rear legs are analyzed. We expect that this difference is partly due to the forward movement of the robot or other potential non-modeled imbalances in the mechanics of the robot. This result is an example of how measuring $I_{CENT}$ can

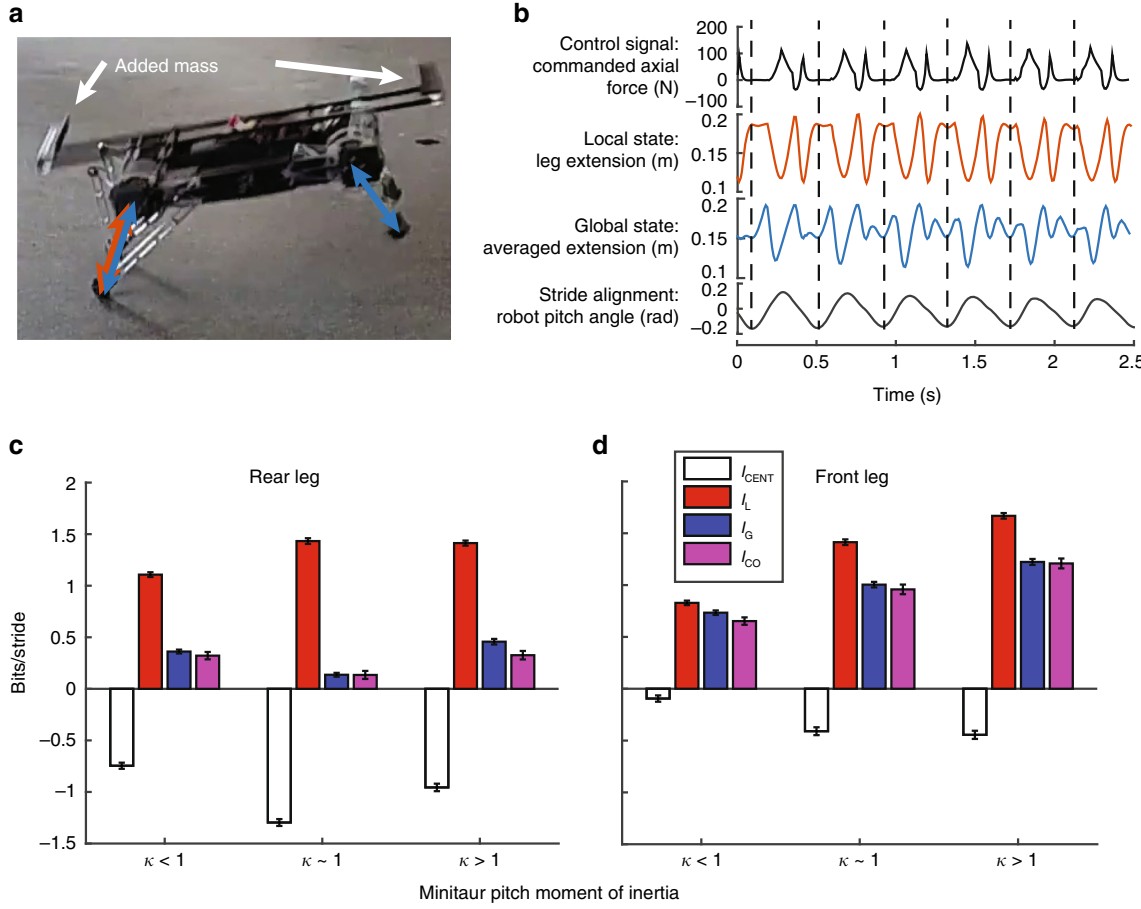

**Fig. 5** Centralization of the bounding Minitaur robot. **a** Image of Minitaur showing the adjustable weight system. The arrows indicate leg extensions used for the local and global states. **b** The commanded torque at the hip to drive the extension of a single leg was selected as the control signal. The local state was the measured extension of that leg. The global state was the average of all leg extension trajectories. Strides were aligned by the pitch angle of the robot. **c** $I_{CENT}$ and $I_{CO}$ of the three different moment of inertia conditions. The estimated axial force to a rear leg was used as the control variable. **d** Estimated axial force to a front leg was used as a control variable

result in discoveries that may not be predicted from simplified models.

Consideration of mechanics is necessary for understanding locomotor control[33]. The virtual leg of running animals all use a similar non-dimensionalized stiffness that also optimizes locomotion in robots[18], allowing a six-legged robot with correctly tuned mechanics to move with just a single actuator[48]. Adding stiff spines to legs[49], flexible joints to the body[50], or streamlined shells[51] allow animals and robots to traverse challenging terrain. The ability to estimate the effects of mechanical feedback such as in these examples could allow for adaptive control[52]. Our centralization measurements detect changes to the mechanics in the robot that might not be evident from kinematics or footfall patterns alone.

**A control architecture space for centralization.** As our centralization measure $I_{CENT}$ does not rely on any model of the underlying dynamics of a system, we can broadly compare where different systems reside in the centralization/decentralization axis. We use a normalization scheme that compares $I_{CENT}$ and $I_{CO}$ in proportion to $I_{TOT}$ shown in Fig. 6 to better compare systems that have different amounts of information depending on capacity. We plot the data from our systems onto these axes in Fig. 6a. We present this representation of centralization and co-information as a control architecture space in Fig. 6b which can be divided

into centralized, decentralized, redundant, and synergistic regions. Comparisons across systems where the dynamics and signals are very different, as is the case with the cockroach and robot systems, should be limited to broad categorizations.

The coupled-oscillator model, which has been used to describe legged locomotion[15] and the control of robots[20], has been used previously to represent gradations of control along the centralized/decentralized axis[2]. When plotted on the $I_{CENT}$ and $I_{CO}$ axes in Fig. 6a, the coupled oscillator does vary from decentralized to centralized as coupling strength increases. However, we also see that fast cockroach locomotion is centralized for the motor unit analyzed, meaning that $I_G$ outweighs $I_L$. The overall $I_{CENT}$ of the cockroach matches that of the coupled-oscillator model with a slightly centralized coupling strength. This result is further validated by a previous study which fit cockroach leg kinematics explicitly to a coupled-oscillator model[12] as indicated in Fig. 6a. Thus we categorize the control architecture of cockroach running as centralized, where coupling in the animal results in proportionally more $I_G$ than $I_L$. In comparison to previous centralization studies in cockroaches we do not have to fit our data explicitly to a specific model (phase-coupled oscillators). Moreover, centralization arises due to information in the timing of muscle activity. The robotic system here is not explicitly a model of the cockroach but rather used to test if our measure captures a hypothesized decentralized architecture and if that

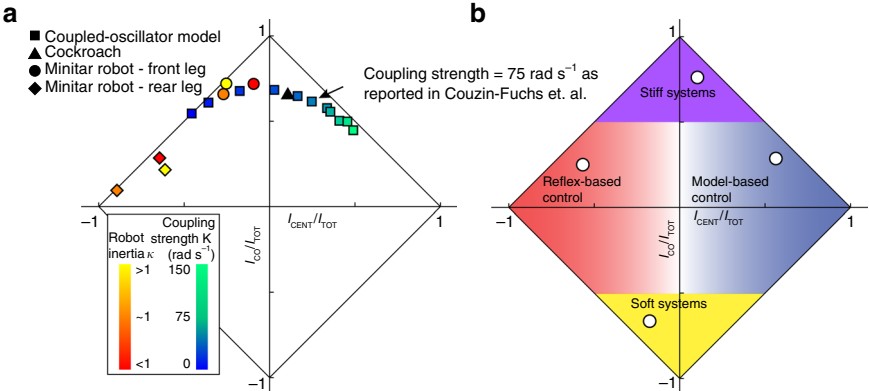

**Fig. 6** Centralization/co-information control architecture space. **a** Centralization and co-information for all systems are plotted here normalized by total information. **b** Any system will fall within a diamond region bounded by the solid black diagonal lines. Centralized systems, where a control signal is more informative about global states than local states, will fall into the blue region. Conversely, decentralized systems will be more weighted toward local information and fall in the red region. The purple region represents systems with high redundancy, where local and global states carry overlapping information about the control signal. The yellow region represents synergistic systems, where knowing both local and global states together with give more information about the control signal than separately. We have indicated where we hypothesize where types of systems might reside

decentralization becomes more pronounced with mechanical decoupling. As predicted, the centralization of the robot is minimized for the condition where mechanical coupling was weakest and in all cases contrasts with the overall centralization of the cockroaches.

The animal, robotic, and coupled-oscillator systems analyzed here span most of the centralization axis. Other systems likely populate the rest of the control architecture space (Fig. 6b), and exploring benefits of the different regions could be a guide in developing robot control or analyzing animal locomotion. While we found no overall change in centralization with speed during running in cockroaches, $I_L$ and $I_G$ did change (Fig. 3). Changing gait is likely to shift location in the control architecture space. Slower walking gaits in cockroaches[53], stick insects[15], and robots[54] are thought to be more decentralized using local or neighbor-based reflex rules[4]. Even though central pattern generator circuits are still involved they are distributed and typically weakly coupled[55] predicting a more decentralized control strategy as indicated in Fig. 6b. On the other hand, robotic control based on a low-dimensional template model[56] would likely be considered centralized as indicated in Fig. 6b.

Different environments might also demand different control strategies. Tests in robotic models indicate that some amount of decoupling between legs, rather than a single centralized controlled trajectory for all legs, results in increased robustness over more variable terrain[57]. These results would predict a leftward shift along $I_{CENT}$ on rough terrain. Movement in either direction on the centralization axis could simplify control, such as a highly actuated elongated fin that only needs to shift trajectories of several of a hundred fin rays to maneuver[58] might be more decentralized, while few control signals driving the coordination of many muscles might be more centralized[59]. From a design perspective, scenarios where positive $I_{CENT}$ is beneficial suggest that it is more important to sense the global state, whereas scenarios where negative $I_{CENT}$ is beneficial suggest emphasizing local state sensing.

In most all examples shown here, $I_{CO}$ is positive, indicating net redundancy. Changes along the $I_{CO}$ axis are possible and could give different performance benefits. For the example of mechanical coupling, stiffer legs would likely result in a highly redundant system in the purple area in Fig. 6b. In terms of maximizing the possible information the control variable could share with both the local and global states, it would be beneficial to have synergistic information rather than redundant information. Such

a scenario could be possible if the controller receives both global and local feedback, where both types of feedback affect the control signal differently together than they do separately, i.e., the contributions from both sources do not simply sum. We expect soft animals and robots without skeletons could benefit from a synergistic control architecture (the yellow region in Fig. 6) because local states might be very independent from global states. One example is a robotic slime mold where each actuator on the edge receives feedback relating to its local neighbors as well as the inner protoplasm that globally interacts with all actuators[60]. The controller takes advantage of the different information the global and local state provides, which would suggest synergy and predict a location in the bottom half of the control architecture spaces.

## Discussion

Our centralization measure is validated against a coupled-oscillator model of centralization in cockroaches, answers an outstanding biological hypothesis about how centralization is maintained across cockroach running speeds even as the local and global information change, and finally tests a hypothesis about mechanical decentralization in a legged robotic platform. Of course, the interpretation of these results are dependent on the chosen representation of the control, local, and global signals. Nonetheless, the comparison of the same system with a varying parameter can be done quantitatively, as we do for the coupling strength in the oscillator model, the speed in the cockroach, and the mechanical coupling in the robot. Also, the model-free nature of the centralization and co-information measures allows comparisons in the categorization of differing systems in the control architecture space.

Finally, while analyzing various systems using our measure of centralization is useful for testing hypotheses or quantifying the degree of centralization, it could also be used as a basis for improving control. First of all, mutual information and entropy can be estimated, and a controller could detect shifts in centralization as the environment changes or body parameters unexpectedly shift due to loading. The controller could then alter coupling to return the system to the desired centralization, either through morphological changes or adjustments to control parameters. Furthermore, centralization could be controlled variably at different hierarchical levels in more complex control architectures. Quantifying centralization with an information-theoretic formulation has the potential to facilitate analysis of a large range

of complex systems beyond the locomotor systems to swarms, networks, or logistics, where similar concepts of centralization versus decentralization are important.

## Methods

**Cockroach experiments**. *Blaberus discoidalis* (henceforth cockroaches) were kept in an incubation chamber set to 37 °C, 60% humidity, and a 12 h/12 h light cycle with ample supply to food and water. Cockroaches were first cold anesthetized in a refrigerator at 4 °C for about 30 min. We then removed their wings and cut back their pronotum so that their legs would be more visible for our overhead video recordings.

To insert the EMG wires, we first restrained them ventral side up to gain access to their legs. The waxy coating on their abdomen and legs was scored with an insect pin to provide better adhesion for the super glue. We made a pair of small holes about two millimeters apart through the exoskeleton of their medial coxa on both the left mesothoracic and metathoracic legs to gain access to femoral extensor muscles 137 and 179, respectively. We then inserted insulated silver wire electrodes (0.003 in wire diameter, A-M Systems, Sequim, WA) into the holes just underneath the exoskeleton and glued them in place. A fifth ground electrode was inserted and glued into the abdomen following the same procedure. The wires were routed along the abdomen and glued on one rostral and one caudal segment. The light tether trailed behind the cockroach and was elevated to a connector above the experimental chamber. These methods are similar to those in refs. [7,61].

Each electrode pair was amplified 100× using a differential amplifier (A-M Systems, Sequim, WA). Amplified signals were recorded through a data acquisition board (National Instruments, Austin, TX) and logged using custom software written in Matlab (Mathworks, Natick, MA). High-speed video (Photron, San Diego, CA) was recorded at 500 fps from above. The arena was lighted with an array of infrared LEDs (Larson Electronics LLC, Kemp, TX). We prodded the cockroaches to run through a narrow opening that led to a wider field and recorded only trials in which the cockroach remained at least a centimeter from the walls. After 12 successful trials each lasting <2 s containing 5–20 strides each, videos were downloaded from the camera to a hard drive, and the cockroach rested for around 10 min until the next set of 12 trials. Up to 8 sets of 12 trials were collected per individual.

EMG data were processed offline using a digital bandpass filter. A simple peak finding method was used to discriminate spikes from the filtered EMG data. The 2D kinematics of the endpoints of all six legs were tracked semi-automatically in the horizontal plane from the high-speed video using custom software written in Matlab. Cubic spline interpolation was used to estimate the position of the leg endpoints during occlusions. Interpolated kinematics were manually checked for a subset of videos to insure accuracy. A global phase variable was estimated using the Phaser algorithm[62] and subsequently used to separate both the EMG and leg kinematic data into individual strides. Stride frequency was estimated from the average change in global phase versus change in time over a stride.

**Robot experiments**. The Minitaur robot was commanded to bound as described in ref. [47], while the translation and orientation of the robot was controlled remotely by a human operator. Runs consisted of around 30–60 s of continuous forward locomotion. Three inertial conditions were tested as described in the main text. For the condition where $\kappa < 1$, the robot was operated without any additional weight. Two 0.5 kg weights were attached symmetrically as shown in Fig. 5a. For the condition where $\kappa > 1$, the weights were placed about 0.35 m from the center of mass. For the condition where $\kappa \approx 1$, the weights were place about 0.2 m from the center of mass. Data from the inertial measurement unit, motor encoders, and estimated motor torques were continuously logged at 100 Hz to on-board memory for later offline processing.

**Estimating mutual information**. We used the $k$-nearest neighbor method for estimating mutual information[63]. In brief, underlying probability densities are estimated by finding the distance to the $k$-nearest neighbor of each sample point, where smaller distances mean the probability of getting that sample is higher. Entropy for the marginal and joint distributions can be calculated from these estimated probability densities, and mutual information can then be calculated. The algorithm in ref. [63] sets the distance to the $k$-nearest neighbor in the joint distribution space, and then uses that distance to find the $k$ for the marginal spaces by counting the samples that fall within that distance. Two slightly different ways of counting these samples are given for estimating mutual information, yet they had little difference in the estimate of mutual information for our data. The choice of $k$ in the joint distribution space sets the resolution to which the probability densities are estimated as the method assumes a uniform distribution in the ball smaller than the distance to the $k$-nearest neighbor. For further details of the estimator see ref. [63].

As a brief aside, the entropy estimator underlying the mutual information estimation is for the continuous form of differential entropy, which is different from the discrete form of entropy described above. However, mutual information has the same properties whether calculated from discrete or differential entropy. Thus, this estimate works for both continous and discrete variables.

We renormalized our variables to have zero mean and unit variance. Such a reparametrization has no impact on the actual mutual information between two variables, but can produce a better estimate as each variable is scaled equally and outliers have a smaller influence[63]. Also, as our the number of spikes is discrete, we added a small amount of noise with a standard deviation of $10^{-4}$ as otherwise many points in the data set would have the same coordinates and therefore counting to the $k$-nearest neighbor becomes impractical as was done in ref. [36]. So long as this noise is small, the amount of noise does not affect the mutual information estimates.

We chose a value of $k$ for which the relative estimates of the different mutual information values remained similar as $k$ varied. From Supplementary Fig. 3, values of $k$ between 5 and 10 give the same estimates for count (top plot), and they fall off at the same rate for timing (bottom plot). Because local and global estimates are either constant or change at the same rate with $k$, these estimates give similar values for centralization whether or not normalized by the total information. We therefore use a value of $k = 7$ for calculating centralization and note that conclusions do not depend on changing $k$ between 5 and 10.

We followed a procedure similar to that in ref. [36] to determine the error of our estimate of mutual information. We sub sampled the data into $m$ equally sized and independent groups containing $\lfloor N/m \rfloor$ samples, calculated the mutual information for those $m$ groups, and then calculated the standard deviation of those $m$ mutual information estimates. We repeated this process ten times and averaged the standard deviations ($\sigma$) for each value of $m$. As variance is generally proportional to 1/(sample size), we fit these mean standard deviations to log $\sigma^2 = A + \log m$ relationship and estimated $\sigma$ for the original full dataset by setting $m = 1$. The error bars displayed in Supplementary Fig. 3 show these measured and extrapolated $\sigma$ values. We are also able to assess whether there exists sample-sized bias in our mutual information estimates if estimates of mutual information stay within the error bars as $m$ is increased and the sample size is decreased. As shown in the right column of plots in Supplementary Fig. 3, estimates for count (top plot) remain constant as the data is subdivided and estimates for timing (bottom) fall off with the number of groups at the same rate, resulting in similar estimates of centralization whether or not normalized by total information. Therefore, estimates of centralization do not fall outside of the error bars as the data are subsampled, indicating that we have collected sufficient data to justify our conclusions. Similarly, when comparing information values for different groups, the values are seen as different if they fall outside of the error range of the estimates.

## Data availability
All relevant data are available on Dryad[64]. The provisional DOI is https://doi.org/10.5061/dryad.4vk610r.

## Code availability
Code used to analyze the data is available from the authors.

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

## Acknowledgements

We thank Avik De and Dan Koditschek for facilitating experiments with Minitaur. Shai Revzen, Ilya Nemenman, Dan Goldman, Sam Burden, and Bob Full provided helpful discussions. This work was supported by NSF CAREER MPS/PoLS 1554790 to S.S.

## Author contributions

I.D.N. and S.S. conceived the research. I.D.N. and A.T. performed the experiments. I.D.N. and S.S. wrote the manuscript.

## Additional information

**Competing interests:** The authors declare no competing interests.

