## [Peer Review File · Nature Communications]

Reviewers' comments:

Reviewer #1 (Remarks to the Author):

This paper integrates simulation using a coupled oscillator model, animal experiments, and robot experiments to develop and validate a novel, empirical, model-free measure to quantify the level of centralization in the control of locomotion using an information theoretic approach. This measure would be useful to compare across a broad range of locomotor systems, both biological and artificial. The authors present compelling reasons for the need to define a model-independent, information theory based measure to indicate how centralized the systems are. The study is well designed, the experimental and model results are mostly convincing, and the implications of the results are mostly well thought out. Finally, the authors proposed an information space which spans different control strategies, based on their degree of centralization and redundancy. This framework is used compare the systems discussed in the paper (but see major comment below), and it could potentially be used for designing robot control strategies. Overall, I think that the paper has a good potential to become a piece of landmark work that many future studies can make use of. I am strongly supportive of the work, but I do encourage the authors to make efforts to make some of the claims more vigorous and better thought-out, or be more clear of its limitation and potential pitfalls.

Major comments:

A major issue is that some findings that are central to conclusions could use a bit more careful consideration and explanation.

(1) First, Fig. 1A is a nice summary of control architecture that can shift from centralized to decentralized. However, it may lead readers unfamiliar with locomotor control to think that this itself is a presumed model, which contradicts with the authors' intention to develop a model-free measure of centralization. I wonder if the authors have thought of this and can better explain it. In other words, are there very different control architectures than shown in Fig. 1A (perhaps not in locomotion control?), where the proposed method can still apply (or maybe not)?

(2) In addition, Fig. 6A is a nice summary of the model and experimental results being detailed in the study to show how different kinds of system lie in the control architecture space based on the novel centralization measure. However, as the authors showed in the animal results, how centralized a system is depend on what is being included in the calculation of the mutual information (e.g., spike count alone, spike timing alone, or both). This suggests that where different systems are will depend on how you obtain mutual information in each. And what if muscle length or forces are measured

instead of EMG signals? Then the question is: how meaningful the comparison is between systems, if one obtains different measurements from different systems, given the constraints of each system? Maybe the information values obtained would even change signs from positive to negative and vice versa.

I understand that it would be hard to have unified experimental methods to obtain exactly the same data from different systems. That said, the authors should consider how well the data on the diagram represents an “apple-to-apple” comparison. They should also emphasize that it is important to do this to enable vigorous comparison in discussions so that people using this method in future do not blindly put their data here and draw conclusions without careful consideration of this challenge. This probably could also be mentioned in the abstract.

(3) Finally, results in Figs. S2 and S3, which the conclusions are based on, seem valid, but the authors should better explain and quantify what they mean by “remains consistent” for Fig. S2, what the locations that they chose in the “plateau region” of Fig. S3 means in terms of the robustness of the conclusions. I do not think it would be a major issue even if the robustness or generality is violated in some regions of the parameter space—after all the measure and method uses greatly simplified (collapsing dimension, etc.) and it is already great that they work for most if not all of the parameter space.

Another major issue with the paper is that parts of the results and interpretations are not explained clearly enough, especially for a general readership, especially those not familiar with information theory. I encourage the authors to better do this so that the study and proposed new measure can reach a broader audience and be more easily adopted by others.

(1) This is most apparent in the description and interpretation of results using information theory. The authors discuss quite a bit synergetic information, redundant information, unique information, but they are not defined or explained well enough for non-experts to easily understand. This makes it harder to appreciate the value of the method and experimental findings.

(2) Also, quite a bit of the results and discussions use data about other components of the information not shown, such as I_R and I_{TOT} . It'd help if the authors add these as complete results in supplemental figures.

(3) The other thing is that many subfigures are not referenced in the text, which makes it hard to follow the description of the results (I had to try hard to connect each sentence in results with a subfigure or feature on the figure). Some subfigures are not explained, such as Fig. 3D, E, F. Also,

many figure elements are not well described or not described at all in figure captions or referred to in the text. Doing so would greatly help follow the findings. Also, without clearly referring to them, it is not always clear the conclusions were made based on which part of the data (e.g., results on information from spike count or spike timing or both differ).

(4) Further, some predictions from previously proposed hypotheses about centralization of locomotor control are not well explained before they were stated to be compared to the findings from this study. These should be revised to made more clear.

Finally, I encourage the authors to consider making some effort, if they are not too hard, to better demonstrate the generality of the proposed method.

(1) I encourage the authors to consider is (after consideration of the comment on the differences in data from different systems above) to see if they can put more results from previous studies on to the control architecture space (many of which they discuss nicely in the last paragraph of the paper, Lines 433-438, Lines 444-450, Lines 461-463, Lines 473-479). This would help demonstrate more applicability and generality of their method (although the current one already shows good generality given that a model, an animal, and a robot are already carefully analyzed by this study).

(2) Have the authors thought about whether this can even be applied beyond locomotor systems, but control systems in general? It'd be nice to speculate if they have some insights.

Minor comments:

Title

Should there be a dash in "Information-based"?

Abstract

Line 18: output states—a bit vague without context

Line 24: internal parameters—vague without context

Line 20: remains consistent—not clear exactly what it means

Line 16: When segregated by stride frequency—seems too detailed and unnecessary for an abstract.

Introduction

Line 19: feedback—do you mean neural feedback specifically here?

Line 34: local feedback—do you mean neural feedback specifically here?

Line 28: This hypothesis predicts a reliance more on fast decentralized mechanical and neural responses local to each leg where global information decreases with speed faster than local information (4).

—Do they both decrease with speed because of bandwidth limitation?

Line 40: However, the challenges of measuring centralization in a system, especially without a specific modeling framework, leaves the general questions regarding the varying degree of centralization in control of animal movement largely unresolved.

—The phrase about without framework can be interpreted two ways: 1. You want to develop a measure that does not depend on model (model-free); or 2. There is no model that we can use to do this which makes it challenging. Clarify.

Line 45: potential—seems unnecessary

Line 47: points—nodes?

Fig. 1A:

Can environment be somehow integrated into the schematic? The authors do discuss this at various places. However, if it is too speculative it may be OK not to do so.

It would be useful to describe what the thick and thin connection lines and arrows stand for in caption.

Some of the figure elements are not defined in caption, such as I_UL, I_SYN, etc.

Line 69: The last sentence of this paragraph nicely states how the proposed method is useful for mechanical feedback. It'd be nice to have a sentence at the end of the previous paragraph about neural control as well.

Line 85: Quantifying all of the concepts of centralization so far described rely on a model of the system.—similar to comment above, this can be interpreted two ways: 1. We want to quantify all of these previous concepts, and this requires us to use a model; or 2. All previous methods described requires a model to quantify centralization. Clarify.

Line 86: Is the measure you propose compatible with previous methods? I think the author implied yes by showing briefly (although not highlighted enough) that it is compatible with one previous study using coupled oscillators. If so, this may be useful to highlight better and explicitly state that this is supposed to be compatible with previous methods.

Line 88: conditions—seems a bit vague without context.

Line 90: What unifies concepts of centralization is the amount of global information a control signal shares about the state of the system compared to the amount of local information.

—Somehow I feel this sentence is a bit hard to understand. Shared with what?

Line 93: contexts—seems a bit vague

Line 96: in a quantity—using a quantity?

Line 99: reconstruct changes—reproduce results?

Line 109: that can be used—that here refers to the information space, or centralization and co-information? Clarify.

Line 116: “extension” here seems to refer to an angle, not the action of extending, as it is an example of local states. But this can be confusing. Consider making this more clear. Same comment for Fig. 3A extension.

Line 129: How much can you really infer detailed structure of control architecture using this empirical model-free measure? Maybe be clear that this is on coarse level.

Line 136: joint local and global states—“joint” here seems a bit out of context.

Equation 1. I think this should be better explained. Maybe a more complete review in the supplementary information. It is not intuitive to understand I_{UL} , I_{UG} , I_R , and I_{SYN} , especially when they are negative (it’s also hard to get what I_{CO} means intuitively). Quite a bit of interpretation of the findings depend on understanding them. Can they be somehow visually explained like the other information in Fig. 1A? Similar comment for Eqns. 2 and 3. It is especially hard to understand the paragraph right after Eqn. 3.

Line 144. There seems to be an extra space in I_{SYN} . Similar issue with quite a few of the subscripts, e.g. Equation 3.

Fig. 2B: Is the red trajectory in second panel and local state in third panel that from the red perturbed oscillator?

Fig. 2C: How is each data point here obtained from the trajectories in Fig. 2B? Average over the shaded region? Also there seems to be error bars on the lines. Maybe point it out in caption.

Line 153: It’d help to specify which section of supplementary material.

Line 156: Sentence seems incomplete. Extra space at the beginning of line.

Line 160: I_{co} is lower case, whereas it is higher case elsewhere.

Line 174: missing “that” after “meaning”.

Results

Eqn. 4—Is theta phase? Not defined. Is there a simple physical meaning of the term in the $\sin()$?

Fig. 2—It’d be useful to cite the oscillator model in caption.

Line 194: it’ll help to explain why 10 Hz is chosen.

Line 197: will the level of noise change your conclusions? Is this checked? Same comment for Line 35 in supplementary materials.

Line 198: Should K be italicized?

Line 205: “amplitude” here is a bit confusing as it is for frequency.

Line 211: what does “fully centralized mean”? Do you expect an upper and lower bound without normalizing to I_{TOT} ?

Line 212: I_{CO} matches...—this sentence should also be constrained by “when $K = 0$ ”.

Line 214: Can you better explain why perturbation cannot propagate? Or is it assumed from “fully decentralized”? Sounds valid but a bit unclear.

Line 218: maybe quantify “these high coupling strengths”. Also this sentence is hard to get because I couldn’t understand well all the components of the information, which is mentioned above.

Line 222: missing “that” before “there exist”.

Line 229: “validating” seems OK to say here considering Fig. 6A, but they authors should highlight the validation that their measure gives consistent results as the previous coupled oscillator model’s method. This is not highlighted enough in the coupled oscillator modeling results section.

Line 231: “terrain” seems unnecessary as it’s flat. Maybe “ground”?

Line 236: In the main text, it’d be useful to explain that cockroaches ran at a large range of stride frequencies and speeds so you can test a hypothesis.

Line 239: Count and timing are not defined in the main text, and how they are measured are not explained in supplementary materials. Why is middle leg chosen? Is information calculated using the entire data set with all samples in time?

Line 248: proportion of overall—contribution to overall

Line 252: stabilized—vague

Line 255: is discussion in this paragraph applicable to cockroach only? Insects? Other animals? Clarify.

Line 257: missing space after “scale”.

Line 266: may be useful to specify “in the middle leg” after 135e.

Line 269: extension here means the action of extending and may be confusing given that they mostly refer to a state of extension. Consider clarifying if it doesn’t become too wordy...

Line 272: do you mean control of flexion and reversal, control of start of joint extension, and control of stride length? A bit unclear. It'd help to specify "our results" of what.

Line 273: missing "that" after "indicate".

Line 274: not very clear how "stance and swing" is relevant here. Can you better explain?

Line 278: I_CENT and I_CO in section title seems a bit too technical. Consider a simpler title.

Line 280: is "faster speed" more appropriate than "faster strides" given your hypothesis?

Line 283: need to explain how data are segmented.

Line 288: This difference is due to a similar trend when looking at timing information—it seems better to say that timing is the main cause/contributor of this difference.

Line 294: It'd help to elaborate what this means—bandwidth limitation?

Line 297: is "perhaps" necessary? Is it or is it not?

Line 299: is this based on information/speed or information/stride? Also missing "is" before "closer".

Line 300: I wonder if the authors should (after better reviewing information theory) show all components of information in supplementary figures. Quite a bit of results and discussion make use of information data not shown, like I_TOT, I_R, etc. Maybe this would help understanding the results easier.

Fig. 3: in caption, you state "the timing of the first four spikes". What are the first spikes? Starting from the beginning of trial? Are data in Fig. 1A starting from the beginning of trial? Why not more spikes as as many as 7 are observed?

Fig. 3G seems a bit out of place to the left of Fig. 3E, F, although this is fine if the authors think it is the best arrangement. Did you do statistical tests to verify significant differences? The trends seem clear and unlikely to change any conclusions, but it'd be nice to do this, at least to those that don't differ very much to verify some claims of difference in describing the results.

Fig. 3C: unnecessary space before 0 spike count on the x axis.

Fig. 3D-F: why are Fig. 3E, F only color coded in the range of 0-0.6 time/T? Maybe use a different color map so differences in phase is better highlighted. These panels are also not explained in the main text. In caption, the statement of color coding should apply to both E and F (currently it is only after E; it may be better to move it to after E and F).

Fig. 4. Similar comment about statistics as above for Fig. 3G. What is full and halves? They are not explained. Are the large square markers necessary? They make it harder to see error bars.

Fig. 5A: a bit hard to see blue lines. Also white angle is not defined. Text is a bit small on Fig. 5.

Fig. 5B: considering making last panel y label consistent with that of the animal figure. How are robot trajectories and motor torques measured? Need more details for robot experiments maybe in a new section in supplementary materials.

Fig. 5C, D: need to quantify moment of inertia. How different are they? At least state in the main text.

Line 309: Consider saying that "our results is consistent with the prediction that" rather than the other way around. Also this paragraph is not well organized. Maybe the predictions should come out first clearly from the hypothesis, then elaborate how the results compare.

Line 312: Is "decrease" a verb or noun here? Maybe "while this information is expected to decrease with speed". Also this prediction is hard to understand without better explaining all components of information.

Line 315-323. Are all these previous results for similar conditions (e.g., similar range of speeds, similar control organization e.g., coupled oscillators)? Or is that not critical here? Also at the end of this paragraph it may help to explicitly point out that you found cockroach running to be on the more centralized side.

Line 328: Gait is misspelled.

Line 333: "... and its parts"—may be useful to have a citation.

Line 338: may be useful to specify the translation and rotation is in the sagittal plane (forward/backward, pitching).

Line 344: are they near perfectly out of phase (phase lag is 180 degree)? Maybe useful to specify.

Line 348: is "block" necessary?

Line 354: what is maximal? Is there a theoretical upper/lower bound without normalizing to I_{TOT} ?

Line 349: Can you quantify M , and how they should affect gait based on previous studies (De & Koditschek)? It is not clear how you make predictions without such information, and this makes it harder to understand the meaning of these results.

Line 352: Are torques and extension expected to correlate well? If not, how? How may this affect the information?

Line 366: It sounds a bit like you know beforehand that rear legs are more decentralized, which is not the case. I also couldn't easily find where this prediction is made earlier. For this section, it needs to be more clearly explained how (and why) you expect moment of inertia change the gait and centralization. Maybe do this in the supplementary materials if space is an issue.

Line 373: change "back" to "rear" to be consistent.

Line 379: Could this have to do (at least in part) with asymmetric mass distribution? Have you checked this? If it does, then this should be added to Line 392 as well. If forward movement is key, can you reverse the direction of the robot and see if you get results that support this speculation? Also “new” seems unnecessary for discoveries.

Line 381: The last paragraph of this section is a nice discussion to show the broader use of the method. Could this fit better with the discussion section? If the authors would like to keep it here it is also fine.

Fig. 6: It'd help to give rationale of choosing these two axis for the space in the text. Text is a bit small on the figure.

Fig. 6A: Would it help to show the coupled oscillator model data more continuously rather than a few discrete points, as the authors have a fine sweep simulation? Can the authors add error bars to animal and robot data?

Fig. 6B: would it help to label “more centralized, more decentralized, more synergistic, more redundant” on the figure? Also the 0.5 threshold for the yellow and purple regions is a bit arbitrary. Maybe use of gradually fading color is better.

Line 403: typo in “/-”?

Line 411: Like in many other places, it'd be useful here to refer to the data point on the figure.

Line 412: I don't have a good sense how strong 76.1 is, which makes the number less useful here. What is considered strong and weak? Could this number be easily put into context?

Line 418: this sentence is not clear because how mechanical coupling relates to moment of inertia and change of gait is not clearly explained; see comment above.

Line 426: what do you mean by “explicitly explore”?

Line 432: Reference to Fig. 6 seems unnecessary as you don't show I_L and I_G. Or clarify. Also, it is hard to understand why change of gait will shift location, similar to comment above.

Line 439: different control strategies? Information is just what you use as the measure of centralization of control architecture. Or clarify.

Line 452: missing "to" before "sense".

Line 457: almost all.

Line 462: body coupling via what? Change of moment of inertia or body stiffness? Or more generally? Is this statement only for robots or also for animals?

Line 464: this sentence is also a bit hard to get because information components are not explained well enough.

Line 469: affect the control signal differently together than they do separately—a bit unclear what this means.

Line 470: by "soft", do you mean specifically animals without vertebrae or exoskeleton in their actuated body parts? Those animals can also have soft components in their actuated body parts.

Line 471: refer to Fig. 6B (perhaps after adding labels of "synergistic" etc. Does this synergistic control architecture have anything to do with muscle synergy (Ting & Macpherson)?

References

Refs. 25 and 48 have {} unnecessary. Check all formatting.

The work by Jamsek et al (2010) PhysRevE 81, 036207 (not related to the reviewer) is a potentially relevant reference to be included, since it uses an information theoretic approach (based on mutual information) to detect couplings between oscillators. The work also discusses the methods to identify strength and directionality of couplings, which could be a relevant extension of the method proposed here.

Supplementary materials

Is the information theory reviewed here generalizable to continuous data sets?

In Eqn. S1, is a negative sign missing?

Line 5: should unit be “bit”, not “bits”? Same comment for “bits” in figures.

Line 7: could you briefly define joint and conditional distributions?

It may be better to spell out MI.

Line 30: the k-nearest neighbor method could be better explained. Does “joint” here mean the joint of the animal legs? Or joint distribution mentioned above?

Line 35: do you mean spike count by “spiking variable”? If not, what is it and how is it discrete?

Line 41: “remained consistent” is a bit vague. Same for “consistent” in Line 43.

Line 43: These consistent trends mean that the local and global estimates give consistent values for centralization whether or not normalized by the total information.—Can this be better explained?

Fig. S1: label spike count and spike timing. Also need to define them.

Line 47: Is N/m always an integer?

Line 49: I don't quite understand this sentence about fitting to obtain sigma. Can the fitting be shown on a figure? Why fit this equation? Alternatively, can you not calculate standard deviation directly from the dataset?

Line 51: errorbars should be error bars.

Line 61: a couple millimeters—2 exactly? Also missing "of" after "couple".

Line 62: missing "," before "respectively".

Line 67: x should be the multiplication sign.

Line 70: why is infrared LED used? Is it not visible to the eye?

Line 72: Is 2 seconds enough?

Line 82: * after section title is unnecessary or not denoted.

Line 83: as the data is auto-correlated with time—are you just saying that the data vary with time? If not clarify. Also this expectation that follows could use a reference.

Line 88: can you show some figures to support these findings that conclusions did not change? Same for Line 92.

Fig. S1: dotted should be dashed line in caption. Could this be merged into Fig. 1? It feels a bit duplicate, and the decentralized to centralized schematic here is useful for the main paper.

Fig. S2: need labels for spike count and timing. Consider using A, B, etc. to make figure reference easier. Co-information missing “-”. Consider spelling out MI.

Fig. S3: I wonder what the valley (blue) regions of these data mean for the robustness of the conclusions. Also, what about the robustness when not including count and timing together, but separately? In caption, “maximal” is vague without context.

Is the phase slice method etc. used in animal analyses also used for the robot? Need to elaborate this and other missing details in a robot methods section.

Reviewer #2 (Remarks to the Author):

In this paper, the authors have proposed a model-free measure method of centralization that compares information shared between control signals and both global and local states. The content of this manuscript covers many disciplines, and the methodology proposed is involved with many classical theories and algorithms. The experimental findings will provide a new approach for comparing biological control strategies as well as designing robotic control strategies. In general, the idea is amazing, though some more explanation is needed.

It is very interesting to introduce the information theory for quantitative characterization of the centralization. But it is still not clear about how to calculate these variables, I_L , I_G , etc. In general, the concepts are abstracts and hard to understand for a roboticist. It would be better to explicitly show some example or results in the supporting information (SI). Although the estimation of the mutual information (MI) is given, there is a big gap between the measured control signals and global/local states with the calculated MI. More detailed descriptions are required for easier understanding. I cannot accurately evaluate the potential value of the proposed method without a deep understanding of these backgrounds.

The authors mentioned that the proposed measure of centralization and co-information will guide the designing of robot control. However, at current stage the method can be only used to indicate the importance for considering interactions between the limbs, body and environment when designing control. In general, we know that it is very important, but how to specifically improve the control architecture? Is the proposed method possible to specifically describe the different layers of robot control architecture?

For centralization, the authors conducted experiments on the cockroach during running, whereas a decentralized control is tested it in the robot of bounding gait. Is there any correlation between the two validations? Furthermore, the cockroach is hexapod, but the used Minitaur Robot is a quadruped robot. Why?

There are some minor problems need to be addressed. E.g., coordinates and units of control signal, local state and global state are missing in the Figures 2, 3 and 5. There are some typos, for example, "though" in Line 8, "Gate" in Line 328, duplicated "more" in Lines 434 and 435.

Reviewers' comments:

Reviewer #1 (Remarks to the Author):

This paper integrates simulation using a coupled oscillator model, animal experiments, and robot experiments to develop and validate a novel, empirical, model-free measure to quantify the level of centralization in the control of locomotion using an information theoretic approach. This measure would be useful to compare across a broad range of locomotor systems, both biological and artificial. The authors present compelling reasons for the need to define a model-independent, information theory based measure to indicate how centralized the systems are. The study is well designed, the experimental and model results are mostly convincing, and the implications of the results are mostly well thought out. Finally, the authors proposed an information space which spans different control strategies, based on their degree of centralization and redundancy. This framework is used compare the systems discussed in the paper (but see major comment below), and it could potentially be used for designing robot control strategies. Overall, I think that the paper has a good potential to become a piece of landmark work that many future studies can make use of. I am strongly supportive of the work, but I do encourage the authors to make efforts to make some of the claims more vigorous and better thought-out, or be more clear of its limitation and potential pitfalls.

Major comments:

A major issue is that some findings that are central to conclusions could use a bit more careful consideration and explanation.

(1) First, Fig. 1A is a nice summary of control architecture that can shift from centralized to decentralized. However, it may lead readers unfamiliar with locomotor control to think that this itself is a presumed model, which contradicts with the authors intention to develop a model-free measure of centralization. I wonder if the authors have thought of this and can better explain it. In other words, are there very different control architectures than shown in Fig. 1A (perhaps not in locomotion control?), where the proposed method can still apply (or maybe not)?

***Our centralization measure requires that a system has a trio of variables defined: the control signal, local state, and global state. There are two main points to Fig. 1A. First, it shows an example of a general type of system that has these three signals. This representation works pretty well for the locomotor systems we study in this work, but other representations of different architecture could work with the measure so long as there exists those three signals. Second, Fig. 1A shows the types of coupling that occur in locomotor systems and how changes to those couplings might give rise to more centralized or decentralized systems. This is the main reason we chose this schematic and not a more abstract, simplified schematic. We note that Fig. 1A is not a model in the sense that there is no specific mathematical structure underlying the blocks and connections. We have added more information into the figure 1 caption as well as the introduction to say that this figure is meant to show these two points, and to be specific that model-free means that there is no assumption of the underlying dynamical model connecting the three signals.**

(2) In addition, Fig. 6A is a nice summary of the model and experimental results being detailed in the study to show how different kinds of system lie in the control architecture space based on the novel centralization measure. However, as the authors showed in the

animal results, how centralized a system is depend on what is being included in the calculation of the mutual information (e.g., spike count alone, spike timing alone, or both). This suggests that where different systems are will depend on how you obtain mutual information in each. And what if muscle length or forces are measured instead of EMG signals? Then the questions is: how meaningful the comparison is between systems, if one obtains different measurements from different systems, given the constraints of each system? Maybe the information values obtained would even change signs from positive to negative and vice versa.

I understand that it would be hard to have unified experimental methods to obtain exactly the same data from different systems. That said, the authors should consider how well the data on the diagram represents an “apple-to-apple” comparison. They should also emphasize that it is important to do this to enable vigorous comparison in discussions so that people using this method in future do not blindly put their data here and draw conclusions without careful consideration of this challenge. This probably could also be mentioned in the abstract.

***We appreciate that this being a new way to analyze these systems, care needs to be taken in the conclusions made from this analysis. That is one reason why it was important to analyze the three systems, some of which we had predictions on what the centralization would be, in order to validate the measure. Certainly, different signals will contain different amounts of information, and the centralization may be dependent on how those signals are chosen as the reviewer points out with timing vs. count information. With regards to that point specifically, splitting the information into count and timing was necessary to extract the total information from the EMG signal (as done in Srivistava et. al.), though it also produced interesting results on the relative influence of count vs. timing.**

Regarding the bigger question. First, the design space in Fig. 6A is normalized in an effort to better compare the centralization between systems that might have different amounts of total information. We now emphasize this point at the beginning of the discussion. Second, The interdependencies between the signals are all transformed into the common language of mutual information. Information is the right currency for this comparison because it does not depend on the particular content of the signals. In this way it does not tell us specifics about the dynamics but is appropriately general. This point is now made at the end of the second to last paragraph of the introduction (line 104).

Third, there is still some limitation to how systems can be compared using these information measures (and any measure). Systems with entirely different descriptions of control, local and global states will be more difficult to relate, but as far as we know this is going to true for any measure. Centralization and co-information let us put these representations on to a comparable scale. Certainly the comparison of the same system with different parameters can be done as we do for the coupling strength in the oscillator model, the speed in the cockroach, and the mechanical coupling in the robot. If we compare across systems (like the robot and the cockroach) a difference in the system could because of differences in what is measured, but this is always one hypothesis for why empirical description two systems might differ. We do pick signals that have similarities across the three systems, such as looking at kinematic states and averaging the kinematics to get the global state. We now carefully make this distinction in the introduction around the revised Fig. 1 and in discussion when we discuss comparisons. Last, Fig. 6B is divided into broad categories, which we think will be useful for comparing across systems. For changes within a system, comparisons could be made as we

demonstrate with Fig. 4. We unfortunately cannot fit this point in the tight constraints of the abstract.

(3) Finally, results in Figs. S2 and S3, which the conclusions are based on, seem valid, but the authors should better explain and quantify what they mean by “remains consistent” for Fig. S2, what the locations that they chose in the “plateau region” of Fig. S3 means in terms of the robustness of the conclusions. I do not think it would be a major issue even if the robustness or generality is violated in some regions of the parameter space after all the measure and method uses greatly simplified (collapsing dimension, etc.) and it is already great that they work for most if not all of the parameter space.

***The goal of Figs. S2 and S3 is to show that there are parameters of the estimation that could potentially indicate sample size bias or a lopsided sampling of the variation. We changed the text in the Estimating Mutual Information section in the supplement as follows: ‘Because local and global estimates are either constant or change at the same rate with k , these estimates give consistent values for centralization whether or not normalized by the total information. We therefore use a value of $k = 7$ for calculating centralization and note that conclusions do not depend on changing k between 5 and 10.’**

Change regarding the ‘plateau’: ‘We thus chose two slices that were a half cycle apart that rested on the plateau of both the local and global MI landscape as shown by the black point in Fig.~\ref{slices}. This plateau indicates that the mutual information estimates are robust to moderate changes in the parameters of the dimensionality reduction (i.e. which particular slices are chosen), but also that there are phases in the stride that share less information than others.’

Another major issue with the paper is that parts of the results and interpretations are not explained clearly enough, especially for a general readership, especially those not familiar with information theory. I encourage the authors to better do this so that the study and proposed new measure can reach a broader audience and be more easily adopted by others.

***We thank the reviewer for laying out specific suggestions in how to implement this. As mentioned in the overview to our response we have taken a pass back through the entire paper to address the clarity of explanations. There are changes throughout but we highlight below the responses to each of the reviewer’s specific concerns.**

(1) This is most apparent in the description and interpretation of results using information theory. The authors discuss quite a bit synergetic information, redundant information, unique information, but they are not defined or explained well enough for non-experts to easily understand. This makes it harder to appreciate the value of the method and experimental findings.

***We made large structural changes to the section introducing information theory to better explain what synergistic, redundant, and unique information are, why they are hard to estimate independently (detail in the supplement), and why our measures of centralization and co-information are useful combinations of these components that are easier to estimate. We also expanded the supplement so that interested readers can get more detail behind the theory. We have also added a new supplemental figure (Fig. S2) which shows the information decomposition and recombination into I_{Co} and I_{Cent} .**

There is a very important point here that actually calculating the component information is not straightforward. This is part of the reason why we think information theory has not always been successfully applied especially to the empirical analysis of locomotion and robotic systems. Part of the advantage of I_{CO} and I_{Cent} is that these quantities can be estimated directly from the empirical data using approaches like the k-nearest neighbor methods because they can be broken into combinations of entropies and joint entropies. As a result it is not just that I_{Co} and I_{Cent} , provide a 2D design space instead of a 4D one with all the component information measures, but that the 2D design space is much easier to estimate with real data.

Based on the reviewer's helpful comments we have revised the text to bring this point forward. While it is not critical for the interpretation of our specific results per se, it is a valuable feature of the analysis. See the *An Information Theoretic Measure of Centralization* section in the main text and *Background on Information Theory* section in the supplement.

(2) Also, quite a bit of the results and discussions use data about other components of the information not shown, such as I_R and I_{TOT} . It'd help if the authors add these as complete results in supplemental figures.

***The components of the information are not themselves calculated in part because they are hard to isolate as we described above. Even though a value like I_R is never explicitly calculated, I_{CO} indicates the range of I_R that could be present. The amount of I_R / I_{TOT} can range from I_{CO}/I_{TOT} to where a vertical line would hit the upper boundary of the design space in Fig. 6A. For example, if the point is on the boundary, $I_{CO} = I_R$. This point is now much more clear in the updated information theory section and supplemental figure.**

Unlike the components, I_{TOT} could be included but the values we plot give I_{TOT} according to $I_{TOT} = I_L + I_G - I_{CO}$. we decided to just show those three values with centralization in Figures 2,3, 4, and 5. This is because the total value is not particularly useful for comparison because it can depend on the total capacity of the system and is used just to normalize the design space. We explain this in the first paragraph of the discussion.

(3) The other thing is that many subfigures are not referenced in the text, which makes it hard to follow the description of the results (I had to try hard to connect each sentence in results with a subfigure or feature on the figure). Some subfigures are not explained, such as Fig. 3D, E, F. Also, many figure elements are not well described or not described at all in figure captions or referred to in the text. Doing so would greatly help follow the findings. Also, without clearly referring to them, it is not always clear the conclusions were made based on which part of the data (e.g., results on information from spike count or spike timing or both differ).

***We have updated the text to better reference subfigures and explain all the figure elements, see responses to minor comments for more details. All subfigures are now references in order and we have revised captions to provide more clarity. Fig. 3D-F are now described both in that caption and in the first paragraph of the *Cockroach Centralization During Running* section.**

(4) Further, some predictions from previously proposed hypotheses about centralization of locomotor control are not well explained before they were stated to be compared to the findings from this study. These should be revised to made more clear.

***We have updated the text to better explain these predictions. We revised the background section on centralization in cockroach locomotion in lines 31-33 and the mechanical decentralization of the robot in lines 380-410.**

Finally, I encourage the authors to consider making some effort, if they are not too hard, to better demonstrate the generality of the proposed method.

***We very much appreciate the concrete suggestions below and address the question of generality now at the end of the discussion. As mentioned in the overview to our response we have also made more widespread changes throughout the paper.**

(1) I encourage the authors to consider is (after consideration of the comment on the differences in data from different systems above) to see if they can put more results from previous studies on to the control architecture space (many of which they discuss nicely in the last paragraph of the paper, Lines 433-438, Lines 444-450, Lines 461-463, Lines 473-479). This would help demonstrate more applicability and generality of their method (although the current one already shows good generality given that a model, an animal, and a robot are already carefully analyzed by this study).

*** We cannot exactly place prior results onto our information design space, because the particular values of centralization and co-information would need to be quantified to do so. However, we interpret the reviewer's point to be encouraging us to discuss where other system might lie and to suggest how our information space might help formulate and test hypotheses. To address this, we have added where we might expect different categories of systems to operate in Fig. 6B that compliment the examples given in the text, such as stiff systems being in the redundant section and soft systems being in the synergistic section. Our response to the next point is also related.**

(2) Have the authors thought about whether this can even be applied beyond locomotor systems, but control systems in general? It'd be nice to speculate if they have some insights.

***We've added some thoughts about the more general use of the measure into the end of the discussion (lines 555-561). Along this point, we also include how information measures could be used as observable and controllable aspects of systems (lines 545-555).**

Minor comments:

***Note that some of these changes might be obsolete if the text was reworded after the initial pass through to address these comments**

Title

Should there be a dash in "Information-based"?

***Changed**

Abstract

Line 18: output states - a bit vague without context

***changed to 'between control signal and both local and global states'**

Line 24: internal parameters - vague without context

***changed to '...without any changes to the controller'**

Line 20: remains consistent - not clear exactly what it means

***consistent -> constant**

Line 16: When segregated by stride frequency - seems too detailed and unnecessary for an abstract.

***Removed clause**

Introduction

Line 19: feedback - do you mean neural feedback specifically here?

***changed to sensory feedback**

Line 34: local feedback - do you mean neural feedback specifically here?

***changed to local sensory feedback**

Line 28: This hypothesis predicts a reliance more on fast decentralized mechanical and neural responses local to each leg where global information decreases with speed faster than local information (4).

Do they both decrease with speed because of bandwidth limitation?

***added bandwidth limitations. The hypothesis is that both will decrease, but that global information will decrease before local information. We have clarified in the text.**

Line 40: However, the challenges of measuring centralization in a system, especially without a specific modeling framework, leaves the general questions regarding the varying degree of centralization in control of animal movement largely unresolved.

The phrase about without framework can be interpreted two ways: 1. You want to develop a measure that does not depend on model (model-free); or 2. There is no model that we can use to do this which makes it challenging. Clarify.

*** Both points the reviewer makes are important for why we pursued this measure of centralization. These points are addressed further in the final intro paragraph. However, here we changed the sentence to 'However, there is currently no general measure of centralization for a system that does not rely on a specific modeling framework, leaving questions regarding the varying degree of centralization in control of animal movement largely unresolved.'**

Line 45: potential - seems unnecessary

***removed**

Line 47: points - nodes?

***changed**

Fig. 1A:

Can environment be somehow integrated into the schematic? The authors do discuss this at various places. However, if it is too speculative it may be OK not to do so.

***Environment is an important aspect of the information loop, and it is implicitly in some of these connections (mechanical coupling and feedback may depend on environment). Often it is difficult to explicitly add environment to such diagrams.**

It would be useful to describe what the thick and thin connection lines and arrows stand for in caption.

Some of the figure elements are not defined in caption, such as I_UL, I_SYN, etc.

***Added to caption**

Line 69: The last sentence of this paragraph nicely states how the proposed method is useful for mechanical feedback. It'd be nice to have a sentence at the end of the previous paragraph about neural control as well.

***Added**

Line 85: Quantifying all of the concepts of centralization so far described rely on a model of the system. similar to comment above, this can be interpreted two ways: 1. We want to quantify all of these previous concepts, and this requires us to use a model; or 2. All previous methods described requires a model to quantify centralization. Clarify.

***Changed to better reflect the second point, that previous methods to quantify centralization were model-based and what we propose is empirical and free of any particular model of underlying dynamics connecting control signal to states.**

Line 86: Is the measure you propose compatible with previous methods? I think the author implied yes by showing briefly (although not highlighted enough) that it is compatible with one previous study using coupled oscillators. If so, this may be useful to highlight better and explicitly state that this is supposed to be compatible with previous methods.

***Added that the measure should be in agreement with previous models, specifically the coupled-oscillator model**

Line 88: conditions - seems a bit vague without context.

***Removed conditions, unnecessary here**

Line 90: What unifies concepts of centralization is the amount of global information a control signal shares about the state of the system compared to the amount of local information.

-Somehow I feel this sentence is a bit hard to understand. Shared with what?

***Changed sentence and added a sentence relating the concept back to the figure. See lines 90 - 97**

Line 93: contexts - seems a bit vague

***removed**

Line 96: in a quantity - using a quantity?

***changed**

Line 99: reconstruct changes - reproduce results?

***changed**

Line 109: that can be used - that here refers to the information space, or centralization and co-information? Clarify.

***changed to clarify**

Line 116: 'extension' here seems to refer to an angle, not the action of extending, as it is an example of local states. But this can be confusing. Consider making this more clear. Same comment for Fig. 3A extension.

***changed to longitudinal extension of one leg**

Line 129: How much can you really infer detailed structure of control architecture using this empirical model-free measure? Maybe be clear that this is on coarse level.

***We changed this to 'we aim to broadly measure whether the structure of the control architecture is more centralized or decentralized.'**

Line 136: joint local and global states - 'joint' here seems a bit out of context.

***changed to 'both local and global states together'**

Equation 1. I think this should be better explained. Maybe a more complete review in the supplementary information. It is not intuitive to understand I_{UL} , I_{UG} , I_R , and I_{SYN} , especially when they are negative (it's also hard to get what I_{CO} means intuitively). Quite a bit of interpretation of the findings depend on understanding them. Can they be somehow visually explained like the other information in Fig. 1A? Similar comment for Eqns. 2 and 3. It is especially hard to understand the paragraph right after Eqn. 3.

***we changed how these components are introduced to be more intuitive and clearer in what they represent, as well as how they are embedded in the quantities we can actually measure. This is addressed in more detail in our responses to the reviewer's main points.**

Line 144. There seems to be an extra space in I_{SYN} . Similar issue with quite a few of the subscripts, e.g. Equation 3.

***This was a latex mistake. We have fixed it on our end and will work with the copy editors to make sure this renders correctly in the manuscript.**

Fig. 2B: Is the red trajectory in second panel and local state in third panel that from the red perturbed oscillator?

***Yes, added sentence to caption**

Fig. 2C: How is each data point here obtained from the trajectories in Fig. 2B? Average over the shaded region? Also there seems to be error bars on the lines. Maybe point it out in caption.

***Added these points to the caption**

Line 153: It'd help to specify which section of supplementary material.

***added**

Line 156: Sentence seems incomplete. Extra space at the beginning of line.

***Fixed**

Line 160: I_{co} is lower case, whereas it is higher case elsewhere.

***fixed**

Line 174: missing $\hat{\theta}$ after $\hat{\omega}$.

***fixed**

Results

Eqn. 4 - Is theta phase? Not defined. Is there a simple physical meaning of the term in the $\sin()$?

***Added 'The sinusoid coupling term is zero when phases are at the preferred phase difference and drive the phases towards that phase difference otherwise.' to text**

Fig. 2 – It'd be useful to cite the oscillator model in caption.

***Added reference to the equation**

Line 194: it'll help to explain why 10 Hz is chosen.

***Chosen to be comparable to cockroaches, added to text**

Line 197: will the level of noise change your conclusions? Is this checked? Same comment for Line 35 in supplementary materials.

***Level of noise give same trends. Added this point to the supplement at that point**

Line 198: Should K be italicized?

***fixed**

Line 205: 'amplitude' here is a bit confusing as it is for frequency.

***changed to 'pulse height'**

Line 211: what does 'fully centralized mean'? Do you expect an upper and lower bound without normalizing to I_{TOT} ?

***The reviewer is correct that our measure could be further decentralized according to the rescaling to I_{TOT} , therefore we changed this to 'highly decentralized'. This particular model cannot be more decentralized, but a larger decoupled network would fall further on the decentralized part of the scale.**

Line 212: I_{CO} matches-this sentence should also be constrained by 'when $K = 0$ '.

***Added 'Also,' to beginning of sentence to inform the reader that this is still the condition $K = 0$.**

Line 214: Can you better explain why perturbation cannot propogate? Or is it assumed from 'fully decentralized'? Sounds valid but a bit unclear.

***This is from Eqn 4, the phase of each oscillator will only depend on its own phase due to the coupling term being zero. We updated the text to succinctly reflect this fact (Lines 246-247).**

Line 218: maybe quantify 'these high coupling strengths'. Also this sentence is hard to get because I couldn't understand well all the components of the information, which is mentioned above.

***Quantified to 'above $K = 150$ '**

Line 222: missing 'that' before 'there exist'.

***added**

Line 229: validating seems OK to say here considering Fig. 6A, but they authors should highlight the validation that their measure gives consistent results as the previous coupled oscillator model's method. This is not highlighted enough in the coupled oscillator modeling results section.

***Since this validation requires the cockroach results, we feel that this point should be made there, and we have added a sentence to that results section.**

Line 231: 'terrain' seems unnecessary as it's flat. Maybe 'ground'?

***changed**

Line 236: In the main text, it'd be useful to explain that cockroaches ran at a large range of stride frequencies and speeds so you can test a hypothesis.

***added**

Line 239: Count and timing are not defined in the main text, and how they are measured are not explained in supplementary materials. Why is middle leg chosen? Is information calculated using the entire data set with all samples in time?

***added some references to the figure 3 subfigures in the main text. Also added a bit more explanation to count and timing in both the main text and the supplementary material**

Line 248: proportion of overall - contribution to overall

***changed**

Line 252: stabilized -vague

*** changed to 'A two-dimensional representation of the output states produced consistent mutual information estimates as higher dimensional representations'**

Line 255: is discussion in this paragraph applicable to cockroach only? Insects? Other animals? Clarify.

***clarified – we reference both insects and bird studies**

Line 257: missing space after 'scale'.

***fixed**

Line 266: may be useful to specify 'in the middle leg' after 135e.

***added**

Line 269: extension here means the action of extending and may be confusing given that they mostly refer to a state of extension. Consider clarifying if it doesn't become too wordy...

***We agree this is somewhat confusing but other wordier options didn't seem to be better**

Line 272: do you mean control of flexion and reversal, control of start of joint extension, and control of stride length? A bit unclear. It'd help to specify 'our results' of what.

***Rewrote this section to help clarify**

Line 273: missing 'that' after 'indicate'.

***added**

Line 274: not very clear how 'stance and swing' is relevant here. Can you better explain?

***stance is associated with leg extension and swing is associated with leg flexion, added this clarification.**

Line 278: I_CENT and I_CO in section title seems a bit too technical. Consider a simpler title.

***changed to 'The Effect of Speed on Shared Information and Centralization'**

Line 280: is 'faster speed' more appropriate than 'faster strides' given your hypothesis?

***changed**

Line 283: need to explain how data are segmented.

***segmented simply by a slow half and fast half, updated text**

Line 288: This difference is due to a similar trend when looking at timing information. It seems better to say that timing is the main cause/contributor of this difference.

***changed**

Line 294: It'd help to elaborate what this means bandwidth limitation?

***This is somewhat elaborated on in the last paragraph of this section**

Line 297: is 'perhaps' necessary? Is it or is it not?

***This is supposed to set up a question that this paragraph tries to answer**

Line 299: is this based on information/speed or information/stride? Also missing 'is' before 'closer'.

***added 'per stride'**

Line 300: I wonder if the authors should (after better reviewing information theory) show all components of information in supplementary figures. Quite a bit of results and discussion make use of information data not shown, like I_{TOT} , I_R , etc. Maybe this would help understanding the results easier.

***This is addressed now in the revision to major comments, but in recap I_R is not directly measured, though a range can be placed on it, as we now explain better in the text. I_{TOT} is $I_L + I_G - I_{CO}$ so we chose not to include it as it is easily calculated from the plots**

Fig. 3: in caption, you state 'the timing of the first four spikes'. What are the first spikes? Starting from the beginning of trial? Are data in Fig. 1A starting from the beginning of trial? Why not more spikes as as many as 7 are observed?

***timing of spikes are by stride, not by the beginning of trial, updated caption to clarify. 1A is taken at a random point of a random trial. As events where more than 5 spikes are rare, the distributions are sparsely sampled and these events do not contribute much to the overall information.**

Fig. 3G seems a bit out of place to the left of Fig. 3E, F, although this is fine if the authors thinks it is the best arrangement. Did you do statistical tests to verify significant differences? The trends seem clear and unlikely to change any conclusions, but it'd be nice to do this, at least to those that don't differ very much to verify some claims of difference in describing the results.

***This arrangement allowed for the time axes of D,E,and F to line up, which we think is helpful and why G is to the left. Statistical tests on information measures is difficult because even though we have reported a standard deviation of the estimates because there is no correct view of the degrees of freedom. Naively, we could use the number of animals - 1 (8) as our degrees of freedom, and the values that we say are different or different than zeros would have p-values less than 0.05. We do not think that approach is entirely accurate, so we avoid using it in this analysis. We instead use the error of the estimate to say that when two values that fall outside of those error ranges have been considered different, as has been done in literature which we now mention in the supplement (Srivastava et al 2017).**

Fig. 3C: unnecessary space before 0 spike count on the x axis.

***removed white space**

Fig. 3D-F: why are Fig. 3E, F only color coded in the range of 0-0.6 time/T? Maybe use a different color map so differences in phase is better highlighted. These panels are also not explained in the main text. In caption, the statement of color coding should apply to both E and F (currently it is only after E; it may be better to move it to after E and F).

***We currently state that F is colored similarly as in E. The colormap only covers the range in timing of the 1st spike, which is why it is only part of the stride. These panels are now mentioned in the main text**

Fig. 4. Similar comment about statistics as above for Fig. 3G. What is full and halves? They are not explained. Are the large square markers necessary? They make it harder to see error bars.

***We shrunk the size of the markers and now explain full and half in the caption. Again, statistical are assessed as in the response above.**

Fig. 5A: a bit hard to see blue lines. Also white angle is not defined. Text is a bit small on Fig. 5.

***We've updated this figure to be more clear**

Fig. 5B: considering making last panel y label consistent with that of the animal figure. How are robot trajectories and motor torques measured? Need more details for robot experiments maybe in a new section in supplementary materials.

***We've added a section in SI about the robot experiments**

Fig. 5C, D: need to quantify moment of inertia. How different are they? At least state in the main text.

***This is now more quantified**

Line 309: Consider saying that our results is consistent with the prediction that δ rather than the other way around. Also this paragraph is not well organized. Maybe the predictions should come out first clearly from the hypothesis, then elaborate how the results compare.

***Rewrote the start of this paragraph to indicate the prediction first and then what the result bears on that prediction**

Line 312: Is 'decrease' a verb or noun here? Maybe 'while this information is expected to decrease with speed'. Also this prediction is hard to understand without better explaining all components of information.

***rewrote so the word 'decrease' is more clear -> '... while the decrease in information with speed is expected,...' This prediction is solely about the local and global information decreases. The prediction of centralization is about the relative decrease of local vs. global, which has opposing hypotheses and ambiguous evidence in the literature as we discuss.**

Line 315-323. Are all these previous results for similar conditions (e.g., similar range of speeds, similar control organization e.g., coupled oscillators)? Or is that not critical here? Also at the end of this paragraph it may help to explicitly point out that you found cockroach running to be on the more centralized side.

***there are a lot of differences between all of these previous results, though they are all from insect locomotion. We see decrease in information per stride but slight increase in information rate, yet coupling remains unchanged. We added the fact that the cockroach is overall centralized according to our measure in line XXXX.**

Line 328: Gait is misspelled.

***Fixed**

Line 333: '... and its parts' may be useful to have a citation.

***added reference**

Line 338: may be useful to specify the translation and rotation is in the sagittal plane (forward/backward, pitching).

***added**

Line 344: are they near perfectly out of phase (phase lag is 180 degree)? Maybe useful to specify.

***added 'antiphase'**

Line 348: is 'block' necessary?

***removed**

Line 354: what is maximal? Is there a theoretical upper/lower bound without normalizing to I_{TOT} ?

***Maximum I_{CO} is the lower of I_L and I_G , therefore we predict it to be close to I_G if the robot is decentralized and redundant. Updated the text to reflect this instead of just 'maximum'**

Line 349: Can you quantify M , and how they should affect gait based on previous studies (De & Koditschek)? It is not clear how you make predictions without such information, and this makes it harder to understand the meaning of these results.

***Rewrote a better explanation of how moment of inertia affects gait and added more quantification, also made the terminology less confusing.**

Line 352: Are torques and extension expected to correlate well? If not, how? How may this affect the information?

***Torque trajectories can indeed be very different from extension trajectories. We chose kinematic states to be consistent with previous work modeling the coupling between legs using kinematics (Couzin-Fuchs et al.) It would be interesting to see what differences might exist if the experiments were repeated with single leg ground reaction forces versus whole body forces.**

Line 366: It sounds a bit like you know beforehand that rear legs are more decentralized, which is not the case. I also couldn't easily find where this prediction is made earlier. For this section, it needs to be more clearly explained how (and why) you expect moment of inertia change the gait and centralization. Maybe do this in the supplementary materials if space is an issue.

***Removed 'more decentralized', we also updated the text to be more clear about how moment of inertia affects centralization**

Line 373: change 'back' to 'rear' to be consistent.

***changed**

Line 379: Could this have to do (at least in part) with asymmetric mass distribution? Have you checked this? If it does, then this should be added to Line 392 as well. If forward movement is key, can you reverse the direction of the robot and see if you get results that support this speculation? Also 'new' seems unnecessary for discoveries.

***There is the possibility that asymmetries in the robot may contribute to the difference in centralization between the rear and front leg pairs. Experiments to further investigate the mechanical coupling of the robot are out of the scope of this paper, but something we are considering in our ongoing work. Removed 'new'.**

Line 381: The last paragraph of this section is a nice discussion to show the broader use of the method. Could this fit better with the discussion section? If the authors would like to keep it here it is also fine.

***We chose to include this paragraph here as a specific reason for needing this method with regards to mechanical coupling considerations, as that is the main focus of the robotic experiment.**

Fig. 6: It'd help to give rationale of choosing these two axis for the space in the text. Text is a bit small on the figure.

***Increased text size and figure size, we think the redone information theory section makes more sense of why these two axes are chosen.**

Fig. 6A: Would it help to show the coupled oscillator model data more continuously rather than a few discrete points, as the authors have a fine sweep simulation? Can the authors add error bars to animal and robot data?

***The data points here are the actual points simulated, we tried connecting them but they made the plot too cluttered. Similarly, adding errorbars in both axes would clutter up the plot, and these errorbars are shown in the previous figures better where specific quantitative comparisons are made.**

Fig. 6B: would it help to label 'more centralized, more decentralized, more synergistic, more redundant' on the figure? Also the 0.5 threshold for the yellow and purple regions is a bit arbitrary. Maybe use of gradually fading color is better.

***We have added hypothesized categories of systems that could go in the plot as was suggested by the major comment. We agree the division of this plot is a bit arbitrary, we experimented with other ways to color the plot including the gradually fading color in both directions but it didn't look as clear as the current figure. We have added a note that this cut off is somewhat arbitrary.**

Line 403: typo in '/'-?'

***fixed**

Line 411: Like in many other places, it'd be useful here to refer to the data point on the figure.

***added in reference to fig.**

Line 412: I don't have a good sense how strong 76.1 is, which makes the number less useful here. What is considered strong and weak? Could this number be easily put into context?

***Part of the issue with getting coupling strengths is that there is little intuition behind the actual numbers, which is another reason why we think centralization is helpful.**

Line 418: this sentence is not clear because how mechanical coupling relates to moment of inertia and change of gait is not clearly explained; see comment above.

*** this is more clearly explained now**

Line 426: what do you mean by 'explicitly explore' ?

***changed to 'span most of the...'**

Line 432: Reference to Fig. 6 seems unnecessary as you don't show I_L and I_G . Or clarify. Also, it is hard to understand why change of gait will shift location, similar to comment above.

***meant to reference Fig. 3, updated. Changing to slow gaits is hypothesized to be more decentralized in the papers referenced**

Line 439: different control strategies? Information is just what you use as the measure of centralization of control architecture. Or clarify.

***changed to control**

Line 452: missing 'to' before 'sense'.

***added**

Line 457: almost all.

***changed**

Line 462: body coupling via what? Change of moment of inertia or body stiffness? Or more generally? Is this statement only for robots or also for animals?

***This is supposed to be a general statement about stiffness, deleted 'body coupling'**

Line 464: this sentence is also a bit hard to get because information components are not explained well enough.

***Revised**

Line 469: affect the control signal differently together than they do separately - bit unclear what this means.

***added ' i.e. the contributions from both sources do not simply sum'**

Line 470: by 'soft', do you mean specifically animals without vertebrae or exoskeleton in their actuated body parts? Those animals can also have soft components in their actuated body parts.

***added 'without skeletons'**

Line 471: refer to Fig. 6B (perhaps after adding labels of 'synergistic' etc. Does this synergistic control architecture have anything to do with muscle synergy (Ting & Macpherson)?

***added reference to figure. Muscle synergies are different than the synergistic local and global information. Muscle synergies are likely more to be representative of redundant information between muscles, though that is not the focus of the paper and we realized that the terminology might be confusing. We added a note about this to the supplement line 67.**

References

Refs. 25 and 48 have {} unnecessary. Check all formatting.

***fixed**

The work by Jamsek et al (2010) PhysRevE 81, 036207 (not related to the reviewer) is a potentially relevant reference to be included, since it uses an information theoretic approach (based on mutual information) to detect couplings between oscillators. The work also discusses the methods to identify strength and directionality of couplings, which could be a relevant extension of the method proposed here.

***Added this reference to the end of the coupled oscillator results section. This is interesting work using information theory to analyze coupling strength and direction in oscillator networks using similar information theoretic tools. Our work looks at a particular aspect of the models, the centralization, and applies it to locomotor systems, though the techniques presented in this work could also be used to learn other new aspects of locomotor systems.**

Supplementary materials

Is the information theory reviewed here generalizable to continuous data sets?

***There is a continuous version of entropy called differential entropy that is not exactly a generalized version of discrete entropy. However, mutual information is the same whether calculated from discrete or differential entropy. The mutual information estimator we use is based off of estimators of differential entropy. We add this point in the SI at line 98**

In Eqn. S1, is a negative sign missing?

***Yes, fixed**

Line 5: should unit be $\hat{\text{bit}}$, not $\hat{\text{bits}}$? Same comment for $\hat{\text{bits}}$ in figures.

***We believe 'bits' is the correct plural, same as 'meters', however, we changed this instance to 'when the base of the logarithm is 2, the unit of entropy is the bit'**

Line 7: could you briefly define joint and conditional distributions?

***added brief definitions**

It may be better to spell out MI.

***changed**

Line 30: the k-nearest neighbor method could be better explained. Does 'joint' here mean the joint of the animal legs? Or joint distribution mentioned above?

***added 'joint distribution' We expanded this explanation.**

Line 35: do you mean spike count by 'spiking variable'? If not, what is it and how is it discrete?

***Yes we mean spike count, updated**

Line 41: 'remained consistent' is a bit vague. Same for 'consistent' in Line 43.

***updated**

Line 43: These consistent trends mean that the local and global estimates give consistent values for centralization whether or not normalized by the total information.-Can this be better explained?

***Changed explanation for clarity**

Fig. S1: label spike count and spike timing. Also need to define them.

***Added labels, and added definitions for count and timing in the text under 'dimensionality reduction**

Line 47: Is N/m always an integer?

***we actually used $\text{floor}(N/m)$, updated in text**

Line 49: I don't quite understand this sentence about fitting to obtain sigma. Can the fitting be shown on a figure? Why fit this equation? Alternatively, can you not calculate standard deviation directly from the dataset?

***This fit comes from the linear scaling between variance and 1/sample size. As sample size decreases, variance increases, thus we look at this increase and extrapolate back to the full sample size.**

This procedure is taken from Srivastava et. al. and is shown in a Figure there. We make the procedure more clear in the text and do not see a Figure showing the fitting as necessary.

Line 51: errorbars should be error bars.
***changed**

Line 61: a couple millimeters -2 exactly? Also missing 'of' after 'couple'.
***changed to 'about 2'**

Line 62: missing ',' before 'respectively'.
***added**

Line 67: x should be the multiplication sign.
***changed**

Line 70: why is infrared LED used? Is it not visible to the eye?
***No specific reason to use infrared, just the lighting we had for other experiments**

Line 72: Is 2 seconds enough?
***at least 5 strides are needed per run**

Line 82: * after section title is unnecessary or not denoted.
***removed**

Line 83: as the data is auto-correlated with time - are you just saying that the data vary with time? If not clarify. Also this expectation that follows could use a reference.
***We mean that there is a dependence between the current state and the previous state (there are dynamics)**

Line 88: can you show some figures to support these findings that conclusions did not change? Same for Line 92.
***Fig. S2 is the justification that conclusions do not change, as the difference between local and global mutual information, and therefore centralization, remain unchanged. We updated the text to make this point more clear**

Fig. S1: dotted should be dashed line in caption. Could this be merged into Fig. 1? It feels a bit duplicate, and the decentralized to centralized schematic here is useful for the main paper.
***changed dotted to dashed, we had it in Fig. 1 but it made the figure a bit cluttered so we cut it for a supplementary figure.**

Fig. S2: need labels for spike count and timing. Consider using A, B, etc. to make figure reference easier. Co-information missing δ - δ . Consider spelling out MI.
***updated this Fig. as suggested**

Fig. S3: I wonder what the valley (blue) regions of these data mean for the robustness of the conclusions. Also, what about the robustness when not including count and timing together, but separately? In caption, 'maximal' is vague without context.
***The valley regions are just slices that happen to carry less information. We are after the information across the whole stride, thus we choose slices that give the maximal amount of information between the control signal and the output state. The fact that there is a plateau indicates that choosing a particular slice does not give some**

random spike in information. Which slices share information are actually an interesting result in itself and is mentioned as such in the results (lines 323-324).

Is the phase slice method etc. used in animal analyses also used for the robot? Need to elaborate this and other missing details in a robot methods section.

***We added a robot methods section, the analysis is a bit different as mentioned in the main text. We attempted the slice method but found more stable estimation with more information using 1st principle components due to the fact that now both the control and states are continuous signals, unlike the EMG activity which was discrete.**

Reviewer #2 (Remarks to the Author):

In this paper, the authors have proposed a model-free measure method of centralization that compares information shared between control signals and both global and local states. The content of this manuscript covers many disciplines, and the methodology proposed is involved with many classical theories and algorithms. The experimental findings will provide a new approach for comparing biological control strategies as well as designing robotic control strategies. In general, the idea is amazing, though some more explanation is needed. It is very interesting to introduce the information theory for quantitative characterization of the centralization. But it is still not clear about how to calculate these variables, I_L , I_G , etc. In general, the concepts are abstracts and hard to understand for a roboticist. It would be better to explicitly show some example or results in the supporting information (SI).

***We appreciate that the calculation of these information measures may not be straightforward. Indeed, there is a large body of literature about how to estimate these measures (Ince et. al., Nemenman et. al, Phys. Rev. E 2004, Sober et. al 2018). In the text and the supplement, we give the general theory underlying these measures, as well as the specific treatments of the data that we had to do for this paper (for example, how to parameterize the various signals to get a low dimensional representation, where each stride is an observation). We have expanded these sections to provide more detail. For the actual algorithm for estimating mutual information, we point to Kraskov et. al. 2004. We provide a broad overview of the estimator which now contains more detail. Also, we refer to Srivivasta et. al. 2017, which used this algorithm and analyzed mutual information in a similar way.**

We have expanded the SI to have more details on the whole process of collecting data from the animal or robotic system, processing that data using various dimensionality reduction methods, and then using the mutual information estimator on the low dimensional representations of the signals.

Although the estimation of the mutual information (MI) is given, there is a big gap between the measured control signals and global/local states with the calculated MI. More detailed descriptions are required for easier understanding. I cannot accurately evaluate the potential value of the proposed method without a deep understanding of these backgrounds.

***Similar to the last comment, we have expanded on the theory and the methods and make it more clear that the process requires measuring the signals, organizing the signals into observations (such as strides), parameterizing or reducing the dimension of the signals, estimating mutual information, then validating the estimate by checking**

for sample size bias. We now start the supplement with an overview of this process, and also provide more details for how each part in the process is carried out.

The authors mentioned that the proposed measure of centralization and co-information will guide the designing of robot control. However, at current stage the method can be only used to indicate the importance for considering interactions between the limbs, body and environment when designing control. In general, we know that it is very important, but how to specifically improve the control architecture? Is the proposed method possible to specifically describe the different layers of robot control architecture?

***We agree that our method helps assess the interaction between limbs body and environment when designing control. These interactions are non-trivial. Some examples include when these interactions dynamically change such as variable terrain, loading on the robot, soft systems, etc. In addition, our method is not just useful from an engineering design perspective but also from the standpoint of designing experiments that test hypotheses about biological and engineered systems. With the growing interest in experimental robotics, robophysics, and the interface of biological design and robotics (see Ijspeert, Science, 2014; Aguilar et.al., Reports on Progress in Physics, 2016) these kinds of methods that work in the same way across systems are useful for translation.**

However, we do think that it is also possible to improve the control architecture with this method and we thank the reviewer for prompting us to make this point explicitly. One way to do this would be to use the relative importance of global or local information to suggest sensing of one or the other of these states would be more important for coordinating the response of the animal or robot. For example trying to implement control independently at different leg pairs might have unintended consequences in a highly centralized system even if that centralization arose from unintended mechanical coupling.

A second possibility is the potential to design outer loop control that adjusts the centralization of the system to match changing environmental conditions. We are currently assessing the performance consequences of different degrees of centralization and examining ways to do just this.

As mentioned in the overview to our response, we have revised the discussion in response to both reviewers' helpful comments to address this point about applicability and the scenarios where centralization could be helpful. In the discussion, we have added some comments on how a roboticist could use centralization and mutual information both analyze robotic control design and how to augment control using the measures as controllable variables. For these particular examples please refer to lines 545-560.

The idea of assessing multilayered systems is very interesting. While we do not explicitly address how multilayered control could be addressed with this method, such extensions are certainly possible. For example, it might be interesting to investigate how the centralization of control signals from different layers compares across different architectures. We have added this point in line 555.

For centralization, the authors conducted experiments on the cockroach during running, whereas a decentralized control is tested in the robot of bounding gait. Is there any correlation between the two validations? Furthermore, the cockroach is hexapod, but the used Minitaur Robot is a quadruped robot. Why?

***We used the Minitaur not necessarily as a model of cockroach locomotion but of legged locomotion using decentralized control (i.e. the leg pairs control as independent hoppers that then coordinate through the mechanics which can be varied). It is also a useful platform for us due to the direct drive motors, meaning that the leg state is not rigid and so the local state entropy is significant. That said, for many questions in legged locomotion, many underlying principles can be found regardless of the number of legs (e.g. stiffness of low order template models, climbing comparisons between cockroaches and geckos). Therefore, we see the application of our measure on the Minitaur to both show the diversity of systems that can be analyzed while also showing the potential to compare legged systems more generally.**

There are some minor problems need to be addressed. E.g., coordinates and units of control signal, local state and global state are missing in the Figures 2, 3 and 5. There are some typos, for example, 'though' in Line 8, 'Gate' in Line 328, duplicated 'more' in Lines 434 and 435.

***Thanks for catching these. We've fixed the typos. Added the units to Figures 2,3,and 5.**

Reviewers' comments:

Reviewer #1 (Remarks to the Author):

The authors have addressed most of the comments raised by both reviewers.

However, I do still feel that the major comment about the use of the method to generally comparing across systems needs further clarification.

To this point, the authors made three arguments in the response letter.

1. In the second argument, the authors stated that “Information is the right currency for this comparison because it does not depend on the particular content of the signals”.

However, as their results in Fig. 3G exemplified, centralization does depend on the content of the signals. Whether spike count alone, timing alone, or both together, is included in the calculation, changes the sign of centralization. So, depending on which of these the authors uses, their conclusions of which system is more centralized when comparing between the animal and the robot (or any other system) will change. One can imagine this being the case for the robot, or any other system that is compared.

Can the authors better clarify why such a sensitivity does not undermine such general comparisons, or how this is handled?

2. The third argument sounds reasonable, but I could not quite see this made clear in the revision.

Where exactly “in the introduction around the revised Fig. 1 and in discussion when we discuss comparisons”?

If these potential issues do exist, the authors should be more explicit of this limitation and tone down the claim of generality.

Also, I am still wondering if the unbalanced distribution of robot mass may contribute to the observed centralization measure. If it does contribute substantially or even dominate the result, it may change the authors' conclusion on how centralized the robot is.

Can the authors simply try hanging the robot on threads (carefully with plenty of cushion) to see if center of mass is near the geometric center of the body?

I could not find a section in the Supplementary Information on robot experimental procedures; only data analysis is described.

Specific comments:

Fig. 1 and Fig. S1. The three labels of the three dashed regions are not clearly enough. Consider using red and blue for I_L and I_G labels.

Remain "consistent" still sounds vague. Is "similar" a more accurate word?

Fig. 2: I can't find the shade region in the last sentence of caption.

Line 253: EMG appears here for the first time. Consider spelling out.

Fig. 3D-F are very briefly mentioned in the revised text, but it is not clear what they show or mean without reading the caption. Can this be improved? Figure findings should support the ideas clearly in the text, not hidden in captions.

Line 522: "through by" seems like a typo.

Fig. 6 caption: "We've" should be "We have".

Supporting Information

Section A: Having “..” instead of “.” After subsection titles.

Line 10 “The sampling of these time series should be stationary, i.e. the statistics of the signals should be constant over time.” Not clear what this means exactly.

Line 47: I (2) seems like a formula together. Considering moving citation to after “handles these issues well”.

Reviewer #2 (Remarks to the Author):

The authors have addressed the points raised in my previous review, and publication of the manuscript is recommended.

Reviewer #1 (Remarks to the Author):

The authors have addressed most of the comments raised by both reviewers.

However, I do still feel that the major comment about the use of the method to generally comparing across systems needs further clarification.

To this point, the authors made three arguments in the response letter.

1. In the second argument, the authors stated that “Information is the right currency for this comparison because it does not depend on the particular content of the signals”.

However, as their results in Fig. 3G exemplified, centralization does depend on the content of the signals. Whether spike count alone, timing alone, or both together, is included in the calculation, changes the sign of centralization. So, depending on which of these the authors uses, their conclusions of which system is more centralized when comparing between the animal and the robot (or any other system) will change. One can imagine this being the case for the robot, or any other system that is compared.

Can the authors better clarify why such a sensitivity does not undermine such general comparisons, or how this is handled?

*** When we said ‘does not depend on the particular content of the signals,’ we meant the physical properties of the signals (voltage, displacement, torque, etc.) and the specific functional relationship between the signals (linear, model-predictive, etc.). We recognize that different parameters or representations of a signal can carry more or less information, such as the example of count versus timing. For the discrete events of EMG spikes, they can be fully parameterized by count and timing. This was part of our motivation for testing count and timing. Count alone (which many studies have used to represent neural signals) misses important information. We updated the text at line 104 to reflect this argument.**

However, the parameterization of the spike timing is a specific issue and the reviewer expresses a larger concern about how centralization can depend on the signals considered. This is indeed true and we respond further in the next point.

2. The third argument sounds reasonable, but I could not quite see this made clear in the revision.

Where exactly “in the introduction around the revised Fig. 1 and in discussion when we discuss comparisons”?

If these potential issues do exist, the authors should be more explicit of this limitation and tone down the claim of generality.

***We took the reviewer's comment as an opportunity to carefully assess how we were discussing generality and interpretation of the centralization measure throughout the paper. We have made several more precise distinctions and clarification to address this point. We are also now more upfront about the limitations early in the discussion (and in the introduction of the method) and specifically where we can and cannot make comparisons between systems. Most of these revisions reflect the third argument we made in the last response to reviewers. Here are the specific changes:**

In addition to the changes to line 104 mentioned above regarding this point, we have updated much of the discussion. First, when we introduce the architecture space in line 476, we state how systems could be broadly categorized.

Second, while we do think this method can be generally (though carefully) applied, we agree with the reviewer that comparison across systems should be limited in terms of the conclusions that can be made, as there could be many hypotheses untested as to why two systems reside in different locations in Fig. 6B. We have added a paragraph at line 480 detailing the extent that comparisons can be made within systems and between systems. This paragraph also touches on how the measures must be interpreted based on the chosen representative signals.

Third, we do think that some broad comparisons between systems can be made (Fig. 6B). For example if I_{CENT} is negative vs. positive it means that either that more local or global state information is represented in the control signal. Moreover a system with net redundant I_{Co} means that information is shared between local and global state representations (see the paragraph at line 191). Basically, the more comparable the systems are in their representation the more a comparison between them using this information space is valid, but even overall location on the diagram is a useful indicator of properties. We discuss this point at lines 495-502. We discuss the categorization of the cockroach at lines 519-525 and the robot at lines 536-550. These categorizations are limited to the broad regions they fall in the architecture space and we make clear that they should only be interpreted given the signals we measured.

Also, I am still wondering if the unbalanced distribution of robot mass may contribute to the observed centralization measure. If it does contribute substantially or even dominate the result, it may change the authors' conclusion on how centralized the robot is.

Can the authors simply try hanging the robot on threads (carefully with plenty of cushion) to see if center of mass is near the geometric center of the body?

***The distribution of robot mass could very well be a factor for the particular baseline values of centralization and co-information. However, the important result is how centralization changes with changing moment of inertia (i.e. mechanical coupling) and whether that matches with prediction. Unfortunately, the robot does not exist in the state that it did when these experiments were conducted so we cannot easily provide those measurements, although the fact that the robot dynamics empirically match a model of a symmetrical sagittal hopping robot (De et. al. 2018) indicate that it is pretty balanced. We do discuss though the potential reasons for why the two legs give different centralization values in the paragraph starting at line 443, which include the possibility of any asymmetries in the mass.**

I could not find a section in the Supplementary Information on robot experimental procedures; only data analysis is described.

***Thanks for catching this. We have added a section describing more of the robot experimental procedures.**

Specific comments:

Fig. 1 and Fig. S1. The three labels of the three dashed regions are not clearly enough. Consider using red and blue for I_L and I_G labels.

***Added some color to the labels and changed the color of the arrows so that they are easier to see.**

Remain “consistent” still sounds vague. Is “similar” a more accurate word?

***Changed to ‘similar’ or otherwise removed.**

Fig. 2: I can’t find the shade region in the last sentence of caption.

***The shaded regions are small, we’ve decreased the line thickness so that they are more visible.**

Line 253: EMG appears here for the first time. Consider spelling out.

***Now spelled out**

Fig. 3D-F are very briefly mentioned in the revised text, but it is not clear what they show or mean without reading the caption. Can this be improved? Figure findings should support the ideas clearly in the text, not hidden in captions.

***We’ve added additional detail of these subfigures in the text.**

Line 522: “through by” seems like a typo.

***Fixed**

Fig. 6 caption: “We’ve” should be “We have”.

***Fixed**

Supporting Information

Section A: Having “..” instead of “.” After subsection titles.

***Fixed**

Line 10 “The sampling of these time series should be stationary, i.e. the statistics of the signals should be constant over time.” Not clear what this means exactly.

***Added ‘Therefore, the behavior should not be changing so much that the overall distributions of these signals change.’**

Line 47: I (2) seems like a formula together. Considering moving citation to after “handles these issues well”.

***Fixed**

Reviewer #2 (Remarks to the Author):

The authors have addressed the points raised in my previous review, and publication of the manuscript is recommended.

REVIEWERS' COMMENTS:

Reviewer #1 (Remarks to the Author):

The authors have done a great job in addressing all major remaining issues and I recommend publication of their paper.

A few minor comments:

Line 110, missing "as" after "so long".

Line 283, "" in "Fig.'3E-F" should be a space.

Line 546, missing "," after "However".

Figs. 1B and S1, use red/blue colors for both I_L/G and their small arrows.

Dear Reviewers:

Thank you for these final comments, which are addressed below:

Reviewer #1 (Remarks to the Author):

The authors have done a great job in addressing all major remaining issues and I recommend publication of their paper.

A few minor comments:

Line 110, missing "as" after "so long".

***Fixed**

Line 283, "" in "Fig.'3E-F" should be a space.

***Fixed**

Line 546, missing "," after "However".

***Fixed**

Figs. 1B and S1, use red/blue colors for both I_L/G and their small arrows.

***Fixed**